# The slow self-arresting nature of low-frequency earthquakes

Xueting Wei [1,2], Jiankuan Xu[2,3], Yuxiang Liu[1,2] & Xiaofei Chen [2,4 ✉]

Low-frequency earthquakes are a series of recurring small earthquakes that are thought to compose tectonic tremors. Compared with regular earthquakes of the same magnitude, low-frequency earthquakes have longer source durations and smaller stress drops and slip rates. The mechanism that drives their unusual type of stress accumulation and release processes is unknown. Here, we use phase diagrams of rupture dynamics to explore the connection between low-frequency earthquakes and regular earthquakes. By comparing the source parameters of low-frequency earthquakes from 2001 to 2016 in Parkfield, on the San Andreas Fault, with those from numerical simulations, we conclude that low-frequency earthquakes are earthquakes that self-arrest within the rupture patch without any introduced interference. We also explain the scaling property of low-frequency earthquakes. Our findings suggest a framework for fault deformation in which nucleation asperities can release stress through slow self-arrest processes.

[1] School of Earth and Space Sciences, University of Science and Technology of China, 230026 Hefei, China. [2] Department of Earth and Space Sciences, Southern University of Science and Technology, 518055 Shenzhen, China. [3] Academy for Advanced Interdisciplinary Studies, Southern University of Science and Technology, 518055 Shenzhen, China. [4] Shenzhen Key Laboratory of Deep Offshore Oil and Gas Exploration Technology, Southern University of Science and Technology, 518055 Shenzhen, China. ✉email: chenxf@sustech.edu.cn

As the smallest seismic events of the slow earthquake family, low-frequency earthquakes (LFEs) are a series of small repeating earthquakes that are assumed to compose tectonic tremors[1,2]. Both LFEs and tremors mainly occur in subduction zones[1,3–6] and are deemed local accelerated deep slip events induced by underlying slow slip. A single LFE differs from a small regular earthquake due to its long duration but small slip rate and low stress drop[7,8]. Specifically, under the same seismic moment magnitude, the slip rate and stress drop of LFEs are 2–3 orders of magnitude smaller than those of regular earthquakes, while their source durations are much longer[7,8]. The unusual source characteristics of LFEs are not well understood within the framework of regular earthquake studies, challenging our basic assumptions about how faults rupture to release tectonic stress. However, despite the differences between these two catalogues of earthquakes, waveform correlations between LFEs and regular earthquakes demonstrate that both are ruptures caused by slip on faults[2]. Therefore, a central question arises about the possibility that LFEs and regular earthquakes are different manifestations of the same physical process. If LFEs and regular earthquakes can be reproduced from the same physical source framework, it may explain the unusual source parameters of LFEs and shed light on the underlying mechanism of tremor and, further, the processes that promote earthquakes.

In this work, we quantitatively reproduce LFEs and regular earthquakes under the same traditional numerical dynamic source model. The results of all 268 numerical simulations reveal that the frictional properties of faults determine the rupture pattern, and we discover a new type of earthquake that arrests itself within the nucleation zone. To investigate the relation between observed LFEs and this new type of earthquake, we compare the source parameters between simulated earthquakes and LFEs observed from 2001 to 2016 in the Parkfield–Cholame section of the San Andreas Fault (SAF), where tremors and LFEs have been recently observed[9,10]. We also discuss the scaling property of our model and observed slow earthquakes. Recurrence and migration patterns are considered to be the characteristics of multiple LFEs[11,12] and are beyond the scope of this article. Our work indicates that LFEs are possibly a series of slow earthquakes that can self-arrest within the nucleation zone, and the underlying mechanism connects with the origin of regular seismic events.

## Results

**Phase diagram from slip-weakening friction modelling.** We focus on interpreting the source process of a single LFE. We present a source dynamic model suited for both slow earthquakes and regular earthquakes in a single rupture patch and simulate possible rupture types under different physical conditions. We present a three-dimensional model based on the slip-weakening frictional law[13–15], a traditional frictional law for describing the rupture process of a single earthquake. We first explore a new type of rupture in the slip-weakening frictional law and then discuss the potential connection between this new type of rupture and LFEs. For a specified nucleation asperity (the area where the rupture is initially triggered) and a slip-weakening frictional law, elastodynamic theory governs how rupture proceeds[16]. The model involves all stages of earthquake dynamics, including nucleation, propagation and termination. We consider a strike-slip fault segment embedded in elastic media, incorporating a homogeneous patch that promotes slip surrounded by an unbreakable boundary (Supplementary Fig. S1a). Details of the model settings and simulation methods are provided in the Supplementary Information.

To generalize the results to all scales of ruptures, we define the normalized $\hat{T}_e$ and $\hat{D}_c$ parameters[17] as $\hat{T}_e = T_e/T_u$, $\hat{D}_c = D_c/(R_a \cdot$

$\frac{T_u}{\mu})$, where $T_u$ is the breakdown stress drop, $T_e$ is the dynamic stress drop, $D_c$ denotes the critical slip distance, $R_a$ is the effective radius of the nucleation asperity, which is the radius of the assumed area for the elevated stress to trigger rupture, and $\mu$ is the shear modulus of the media (Fig. 1a). For a single simulation, the inputs are the parameters $\hat{T}_e$, $\hat{D}_c$, and $R_a$, the wave speeds and the density of the fault medium. The output is the rupture process, that is, the slip velocity and stress of the calculated area at each moment based on the given parameters.

The pair of parameters $(\hat{T}_e, \hat{D}_c)$ and the initial stress conditions completely determine the style of rupture. We assume that the initial shear stress inside the nucleation patch has reached the shear strength limit $\sigma_u$, while the initial stress outside the nucleation patch is $\sigma_0$ (since $\sigma_r$ can be eliminated during the calculation, the initial stress in the nucleation patch is actually set to be $T_u$ during simulations, while the initial stress outside the nucleation patch is $T_e$). At 0 s, we assign all points in the nucleation patch a stress slightly greater than the shear strength limit so that rupture occurs across the nucleation patch simultaneously. In the simulations, the nucleation asperity is fixed as a circle with a radius of 100 m and a spatial grid size of $ds = 3$ m.

By varying $\hat{T}_e$ and $\hat{D}_c$ from 0 to 1 (almost all cases of fault friction properties), four rupture patterns can be obtained, and we show the results in a phase diagram of dynamic rupture (Fig. 1c). The rupture triggered at 0 s within the area of the nucleation zone (the assumed area for the elevated stress to trigger rupture) propagates according to different friction properties. In Fig. 1b, for the super-shear rupture, $\hat{D}_c$ is 0.4, and $\hat{T}_e$ is 0.8; for the sub-Rayleigh rupture, $\hat{D}_c$ is 0.4, and $\hat{T}_e$ is 0.4; for the self-arresting rupture, $\hat{D}_c$ is 0.35, and $\hat{T}_e$ is 0.1; and for the slow self-arresting rupture, $\hat{D}_c$ is 0.88, and $\hat{T}_e$ is 0.2. Each point in the phase diagram indicates a rupture simulation with the parameters of that point, which determines the resulting rupture style. The most important feature shown in the phase diagram is the delineation of the four areas that correspond to the conditions for super-shear, sub-Rayleigh, self-arresting and slow self-arresting rupture.

**Slow self-arresting rupture.** The phase diagram reveals that four different types of rupture patterns exist. Based on their different rupture termination styles, we divide these four rupture types into two groups: Both sub-Rayleigh and super-shear ruptures are runaway earthquakes, as their rupture fronts spread continuously until encountering any barrier (Fig. 1b). Conversely, the other two rupture types are self-arresting. These two types spontaneously arrest by themselves even in the absence of any barriers (Fig. 1b). The energy released at the initial stage of the seismic nucleation process is insufficient for the rupture front to overcome the yield strength of the surrounding media. Evidence of self-arresting events is observed for repeating earthquakes along the Parkfield section of the SAF[18].

Here, we define slow self-arresting ruptures (SSARs) as a series of slowly SSARs, which correspond to the large-$\hat{D}_c$ portion of the phase diagram (Fig. 1c). Similar to regular self-arresting events, SSARs are also characterized by their spontaneous arrest slip pattern. However, SSARs emanate seismic waves slowly compared with the other SSARs, and for the same rupture patch area, their slip and stress drop are much smaller, similar to the characteristics of LFEs. In particular, SSARs extinguish themselves within the nucleation patch (the patch we set in numerical simulations to initiate earthquakes), and the whole source process is limited within the nucleation asperity (Fig. 1b). Typically, fast rupture occurs along areas of the fault where the frictional resistance decreases with increased slip, which are called slip-weakening asperities. However, the released stress is quite limited

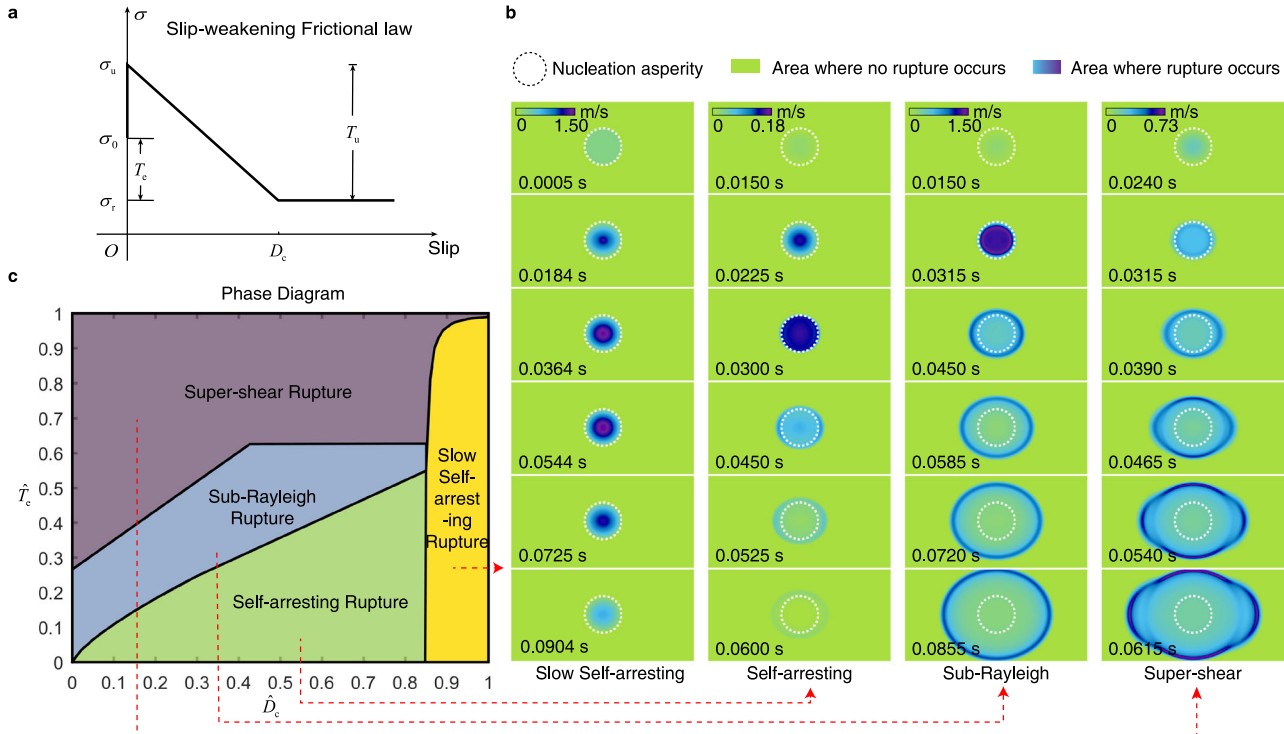

**Fig. 1 Phase diagram of rupture dynamic and four types of rupture modes. a** Slip-weakening frictional law; $\sigma_u$, $\sigma_0$, and $\sigma_r$ are the peak strength, initial stress and residual stress, respectively. $T_u$ is the breakdown stress drop, $T_e$ is the dynamic stress drop. **b** Four types of rupture modes. The simulated strike-slip faults are shown as the green background area, and the white dashed lines enclose the nucleation asperity. All four types of rupture are under the same initial stress state. **c** Rupture phase diagram in parameter spaces ($D_c$, $R_a$, $T_e$, $T_u$) for four different combinations of friction parameters.

if the amount of slip for every point on the fault is much smaller than the critical slip distance $D_c$ (Supplementary Figs. S2, S3). SSARs are similar to the rupture event investigated by previous works[19–21], where the slip cannot reach $D_c$. When the released stress on the rupture is insufficient even to break down the fault media surrounding the nucleation asperity, the rupture process spontaneously self-terminates in the nucleation zone, which is exactly the case for SSARs (Supplementary Fig. S3). A comparison of the source time function of SSARs with those of ordinary self-arresting earthquakes and regular earthquakes (Supplementary Fig. S4) confirms this inference.

**Comparison of LFEs and SSARs.** Inspired by the general similarity between LFEs and SSARs, we compare the simulated SSAR source parameters with specific observed LFEs. The frequent low-frequency seismic activity in the Parkfield area[10] and the existence of a high-precision seismometer network provide a chance to verify the connection between LFEs and SSARs. Using the LFE location data identified by Shelly, we select the LFEs with good correlation with the template (meancc ≥0.4 and ccsum ≥10) to estimate the source parameters. We estimate the average moment magnitudes and stress drops of 6585 LFEs in every tremor family located in the Parkfield from 2002 to 2016 (Fig. 2). The specific estimation methods are provided in the Methods section.

Our results show that the moment magnitudes of LFEs are between 0.7 and 2.2, the stress drops are between $8.1 \times 10^3$ Pa and $1.0 \times 10^5$ Pa, and the average rupture patch diameter is ~240 m (Fig. 2), which are close to the values estimated in previous works (Supplementary Table S1). Combined with the LFE source duration estimated by Thomas et al.[8], we also derive the average slip and slip rate ranges for LFEs. The average moment magnitude of each LFE family is shown in Supplementary Fig. S5. We have also studied the influence of different quality factors and

filter frequency bands on the estimation results (Supplementary Figs. S6–8).

We compare the LFE source parameters derived from the Parkfield area seismic data with those from numerical simulations. We focus on comparing the source characteristics of simulated and observed LFEs on a single rupture patch rather than the overall tremor characteristics. Sub-Rayleigh and SSARs and SSARs are deemed potential source mechanisms for LFEs.

Based on the numerical models described before, we assume that the LFEs emanate from the micro-asperities on the strike-slip fault and that the background dynamic shear stress drop $T_e$ of the Parkfield fault plane is 1 MPa[22,23]. With the average rupture area diameter for LFEs being ~200 m, we simulate all three types of earthquakes with rupture patch diameters set at 180 m, 220 m, 260 m, and 300 m (super-shear ruptures are not relevant). We use four rupture patch diameters during simulations to minimize the impact of the inaccuracy of the estimated LFE average diameter from seismic data on the comparison results. Given the same stress loading condition, for each rupture diameter, we perform simulations with all possible combinations of slip-weakening parameters ($\hat{T}_e$, $\hat{D}_c$) (Supplementary Fig. S1b and Supplementary Table S2) and calculate the source parameters for each case.

The results show that only SSARs satisfy all the requirements of the LFE source parameters at the same time (Fig. 3). Even with the smallest magnitude and the minimum rupture diameter of 180 m, the stress drops and slips of sub-Rayleigh and self-arresting earthquakes still differ significantly from the observed ranges of source parameters for LFEs. This is a notable result because an earthquake with a long source duration commonly results from a large rupture area, moment magnitude and stress drop, rather than a relatively small magnitude and stress drop, which is the case for LFEs. By introducing a new type of earthquake (SSAR) that can arrest itself within the nucleation

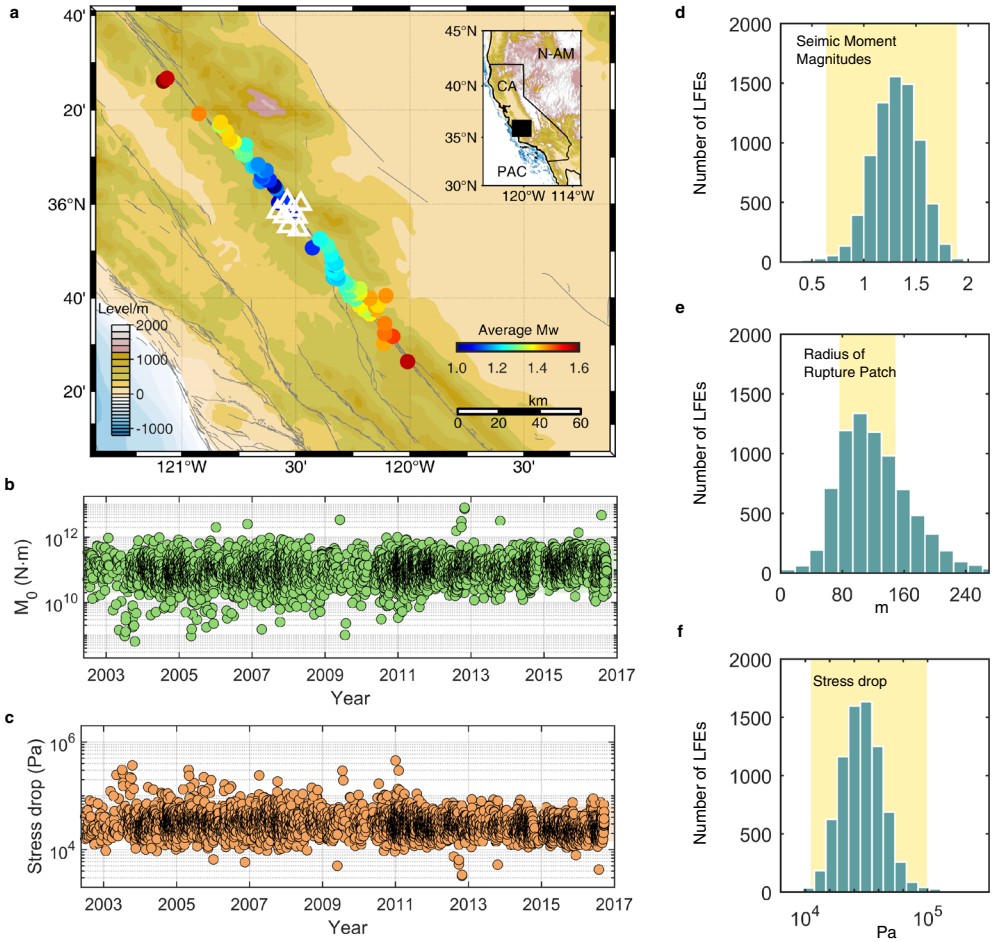

**Fig. 2 Parkfield LFEs. a** Map of the Parkfield, California area, HRSN borehole stations, and associated topography. LFEs identified by Shelly10 are shown as circles, and the HRSN borehole stations are shown as white triangles. Different circle colors represent different average moment magnitudes of LFEs. **b** Seismic moment magnitudes of LFEs in Parkfield (green dots). **c** Stress drop magnitudes of LFEs in Parkfield (orange dots). **d** Histogram of estimated LFE moment magnitudes. The yellow strips show the source parameters range we used in comparison with the simulated results. **e** Histogram of estimated radii of rupture patches of LFEs. **f** Histogram of estimated stress drops of LFEs.

zone, the contradictory characteristics of long duration and small stress drop of LFEs are successfully explained. In addition, the inefficient nature of SSAR energy radiation is revealed, noting that for the same rupture area, the stress drops, slips and slip rates of SSARs are 1–2 orders of magnitude smaller than those of regular earthquakes (sub-Rayleigh ruptures). It is also notable that the ranges of source parameters of the SSARs with large $\hat{T}_e$ values (0.75–0.95) agree well with those of LFEs. The larger $\hat{T}_e$ is, the lower the shear strength limit of a single LFE asperity is (Supplementary Fig. S2), meaning less stress is needed to trigger LFEs. Each SSAR releases only 1–10% of the stress within the rupture patch (fault breakdown stress drop $T_u$); thus, it can quickly return to the initial stress state and rupture again. Promoted by the underlying slow slip events (SSEs), the repeated stress release and accumulation of SSARs may produce tremor-like seismic signals (Supplementary Fig. S9). Therefore, it is reasonable to speculate that the rupture process of an SSAR in a micro-asperity represents the source mechanism of a single LFE.

**Moment-duration scaling property of SSARs.** A controversial topic of slow earthquakes is what scaling law they follow. As noted by Ide et al.[24,25], slow earthquakes may follow a different scaling law from regular earthquakes. However, some evidence shows that slow earthquakes possibly follow a cubic moment-duration scaling law similar to regular earthquakes[26–29]. Here, we

use the SSAR as the source model for LFEs and slow earthquakes and investigate the scaling properties of SSARs (Fig. 4).

Since LFEs are the smallest slow earthquakes and the source duration range of LFEs is very limited (~0.1–1 s)[8], we add more SSAR simulation results other than LFEs simulated before. We assume that LFEs of different scaling are SSARs of different sizes under the same frictional conditions. We simulate SSARs with fixed $\hat{D}_c = 0.9$ and $\hat{T}_e = 0.5$, and the rupture patch radius varies from 200 m to 9 km. We maintain a dynamic stress drop $T_e$ of 1 MPa and the same background velocity structure as in the previous simulations.

We linearly fitted the scaling regulation of all simulated SSARs, and the result can be expressed as:

$$M_0 = T^{3.06} \times 10^{12.98} \tag{1}$$

Our results suggest that the SSARs follow a cubic scaling law that is similar to regular earthquakes and is consistent with some of the studies[26–29]. Since the SSARs/LFEs share the same physical model with regular earthquakes, the similarity of their scaling properties is reasonable. In addition, the duration and seismic moment magnitude range of SSARs with rupture radii of 2–8 km agree well with the observed seismic moment magnitudes of very low-frequency earthquakes (VLFs)[30,31], implying that SSARs can also explain the source characteristics of VLFs. Although the scaling of regular earthquakes and SSARs have similar slopes,

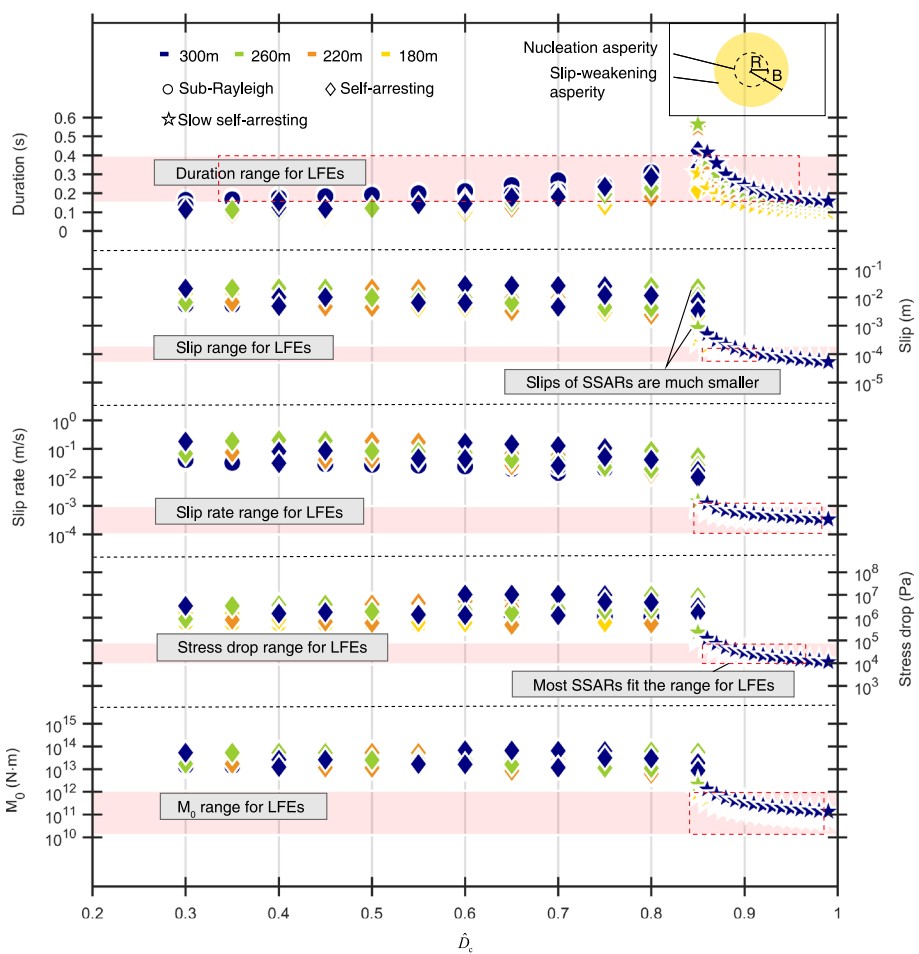

**Fig. 3 Source characteristics comparison between simulated ruptures and LFEs in Parkfield.** The duration, slip, slip rate, stress drop and seismic moment magnitudes are compared. Pink strips denote the estimated source parameters ranges for LFEs in Parkfield, and colored dots show the results of numerical simulated earthquakes. Different dot colors represent different simulated rupture patch diameters, while different dot shapes show different types of ruptures in simulations. The blue, green, orange, and yellow dots denote simulated earthquakes with rupture diameter 300 m, 260 m, 220 m, and 180 m, respectively. The circles, diamonds and stars denote the sub-Rayleigh earthquakes, self-arresting earthquakes, and slow self-arresting earthquakes, respectively. In addition, the red dashed rectangles show the ranges where the source parameters of simulated earthquakes match those of LFEs.

there is a difference in their intercepts. The intercept is caused by the slow nature of SSARs, which corresponds to the large residual stress of SSARs we discussed before. Note also that the similar slope result is based on the assumption that slow earthquakes of different sizes occur under the same frictional conditions with the same $\hat{T}_e$ and $\hat{D}_c$. Therefore, with more observations and experiments uncovering the stress and slip-weakening conditions of LFEs in nature, we may find a more accurate and universal scaling law of LFEs.

## Discussion

Our numerical investigations indicate that the essential difference between regular earthquakes and LFEs is different frictional conditions that drive different ways of rupture termination. While regular earthquakes are terminated by barriers, LFEs gradually self-arrest, which is equivalent to the SSARs in our numerical simulations. The essential factor that controls different ways of rupture arrest is the frictional property of the fault, more specifically, the parameters $\hat{D}_c$ and $\hat{T}_e$, as shown in the phase diagram (Fig. 1c). $\hat{D}_c$ plays an important role in determining the rupture styles. For smaller $\hat{D}_c$ ($\hat{D}_c < 0.85$), the rupture pattern depends on the value of $\hat{T}_e$; however, for larger $\hat{D}_c$ ($\hat{D}_c \geq 0.85$), regardless of the $\hat{T}_e$ magnitude, the rupture is always a slow self-arresting

rupture. Specifically, at $\hat{D}_c = 0.90$, the rupture is an SSAR for all $\hat{T}_e$ values tested (0.01, 0.50, and 0.95). When $\hat{T}_e$ equals 0.95, i.e., when the breakdown stress of the surrounding media is only 5% larger than the background shear stress, SSARs can be easily triggered. Here, we provide a physical model based on our simulation to illustrate how LFEs occur (Supplementary Fig. S10). We assume that both LFEs and normal earthquakes occur in the same area. We control the final rupture area to be 220 m in diameter, $R_a$ of the SSARs is set to 110 m, and $R_a$ of the normal small (run-away) earthquakes is set to 75 m. According to the phase diagram and simulation results, the dimensionless $\hat{T}_e$ range of the LFEs is 0.75–0.95, and that of the normal small earthquakes is 0.3–0.6. The dimensionless $\hat{D}_c$ range of the LFEs is 0.85–0.99, and that of the normal small earthquakes is 0.3–0.75. We choose dimensionless ($\hat{D}_c$, $\hat{T}_e$) as (0.9, 0.85) for the SSARs and (0.5, 0.45) for the normal earthquakes. Therefore, the dimensions ($D_c$, $T_u$) of the SSARs are (3.6 mm, 1.17 MPa), and those of the normal earthquake is (2.6 mm, 2.22 MPa). According to the simulation results, the SSARs are consistent with the source parameter range of the LFEs in Parkfield, while the normal earthquakes are not. The seismic moment magnitude of a normal earthquake is 2.6, which is consistent with the magnitude of normal small earthquakes recorded by the Northern California Earthquake Data Center (NCEDC). The $T_u$ values of normal earthquakes and

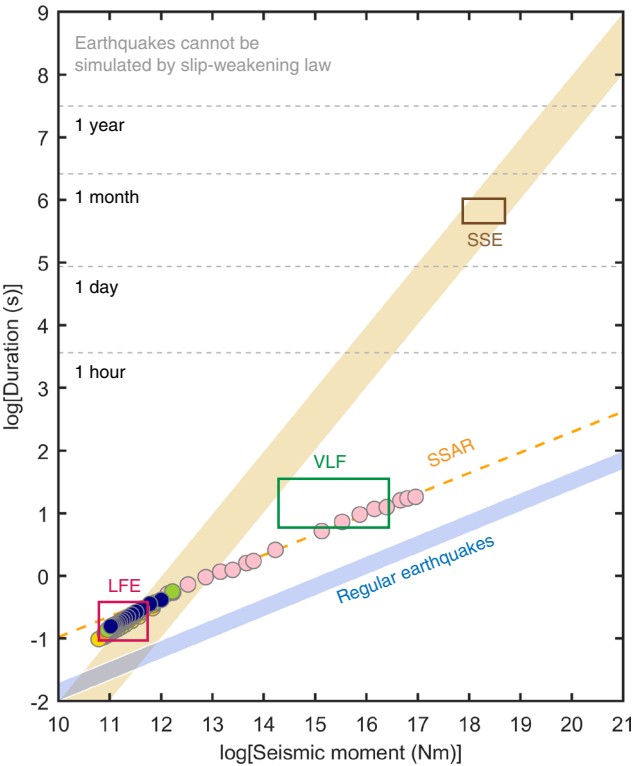

**Fig. 4 Moment-duration scaling relationship of SSARs.** The yellow strip denotes the linear scaling of slow earthquakes reported by Ide et al. 2007[24]. The blue strip denotes the cubic scaling of regular earthquakes. The pink dots denote the SSARs with rupture patch diameters ranging from 200 m to 9 km. The blue, green, orange, and yellow dots show the scaling of SSARs we used before to simulate the LFEs. The blue, green, orange, and yellow dots denote simulated earthquakes with rupture diameter 300 m, 260 m, 220 m, and 180 m, respectively. The orange dashed line shows the fitting result of SSARs. The red rectangle is the area of magnitude and duration consistent with LFEs. The green rectangle is the area of magnitude and duration consistent with VLFs, and the brown rectangle is the area of magnitude and duration consistent with SSEs.

SSARs are very different. For this normal earthquake with $T_u = 2.22$ MPa, an additional triggering stress of 1.22 MPa is required. For the SSAR with $T_u = 1.17$ MPa, an additional triggering stress of only 0.17 MPa is required. This is a plausible scenario, as LFEs are triggered by small external stress disturbances, such as tides or body/surface waves[32–39]. Since $T_e$ corresponds to the fault dynamic shear stress drop, the sensitivity of tremors to shear stress changes becomes understandable. By definition, $\hat{D}_c$ is inversely proportional to the effective radius of the nucleation asperity $R_a$ and the breakdown stress drop $T_u$. This suggests that SSARs/LFEs tend to occur in smaller rupture patches and faults with smaller shear stress, as SSARs occur only in fault media with large $\hat{D}_c$. Unlike previous studies that focused only on multiple characteristics of LFEs[40–42], our model satisfies not only the source characteristics of a single LFE but also the group characteristics of LFEs due to the easy excitation and low energy release characteristics of SSARs.

Equipped with a model that explains the source processes of LFEs, we discuss further implications of our results. According to our simulations, LFEs emerge only in asperities with large normalized critical slip distances $\hat{D}_c$, while SSEs are also reproduced with large slip-weakening distances[43–45]. Since tremors and LFEs are thought to be promoted by the underlying SSEs[46–48], the similarity between the simulation conditions for the two slow seismic phenomena is not coincidental. It seems that a large slip-weakening distance is an

indispensable requirement for slow earthquakes to occur, corresponding to the slow self-arresting process in the phase diagram. This similarity supports our results of a large normalized critical slip distance $\hat{D}_c$ in dynamic modelling, further confirming our inference that LFEs are fundamentally SSARs.

Another unusual characteristic of LFEs reported is that there are some LFEs whose source durations are approximately independent of their seismic moment magnitudes[8,49–51]. This feature may also be explained by SSARs. We simulate SSARs with the same $\hat{D}_c$ value of 0.88, the same rupture patch diameter of 240 m and $\hat{T}_e$ ranging from 0.1 to 0.9. The results show that simulated SSARs have the same source duration but different seismic moments (Supplementary Fig. S11). Their moment magnitudes range from 1.47 to 2.11, and the source durations are both 0.22 s. This is because under the same rupture patch size (nucleation patch size), $\hat{D}_c$ mainly controls the slip rate and propagation method of SSAR, while $\hat{T}_e$ determines the magnitude of absolute shear strength ($T_u$). Under the same $\hat{D}_c$ and rupture patch size, the durations of SSARs are the same, but the moment magnitudes are different due to the difference in $\hat{T}_e$. Therefore, the SSAR is a possible physical model of LFE and explains most of the source features of LFEs.

The tremor forms either through the mutual influence of LFEs on multiple patches or due to LFEs occurring on a single rupture patch multiple times. Since the SSARs represent the source model of LFEs and the rupture area of SSARs is constrained within the patch due to their self-arresting property, we conclude that the LFEs on each rupture patch do not influence other patch rupture processes. Hence, tremors are likely to be composed of multiple LFEs that occur independently of other patches according to our SSAR model.

The rupture phase diagram reveals that small slow and regular earthquakes are rupture events under different frictional conditions. By analyzing different normalized parameters in the phase diagram of rupture dynamics, we discover a new earthquake SSAR that arrests itself within the nucleation zone. Through the comparison between the LFE duration, slip rate, stress drop and moment magnitude from seismic records and the numerical simulation results, we find that only SSARs successfully satisfy the special source parameters of observed LFEs. The scaling law of SSARs follows a cubic moment-duration scaling law, agreeing well with scaling property of slow earthquakes shown by some studies. SSARs, which are characterized by their inefficiency in releasing energy in rupture processes, are easily triggered and have a large normalized critical slip distance, providing a natural explanation for the swarm-like behavior of LFEs. In summary, all results indicate that LFEs are SSARs in nature.

## Methods

**Simulating the LFEs.** Low-frequency earthquakes (LFEs) are a series of small earthquakes that are observed mainly in subduction zones and accompanied by tectonic tremors. The smaller stress drop, moment magnitude, and slip rate and longer source duration distinguish LFEs from regular small earthquakes. Here, we describe the source dynamic model settings for both LFEs and regular earthquakes and the parameters we use in numerical simulations.

**Boundary integral equation method.** There are many methods to solve the spontaneous rupture problem[13,14,52–64]. Among them, the boundary integral equation method (BIEM), as a type of boundary method (another category is domain methods), is more accurate and efficient for a planar fault in homogeneous media. In our study, we use the extended BIEM to simulate fault slip evolution, including nucleation, propagation, and termination.

The extended BIEM of Zhang & Chen[62,63] comes from the representation theorem,

$$u_n(x,t) = \int_{-\infty}^{\infty} d\tau \iint [u_i(\xi,\tau)] c_{ujpq}\nu_j \partial G_{np}(x, t-\tau; \xi, 0)/\partial \xi_q d \quad (2)$$

Substituting Eq. (2) into the isotropic and homogeneous constitutive relationship $\tau_{ij} = \lambda\delta_{ij}\partial_m u_m + \mu(\partial_j u_i + \partial_i u_j)$ and through regularization and discretization, we obtain the following discrete boundary integral equation for

updating the shear stress field on a planar fault:

$$\tau^{ijk} = \tau_0 + \sum_{l,m,n} C^{ijk,lmn} V^{lmn}, \qquad (3)$$

where $\tau^{ijk}$, $\tau_0$ and $V^{lmn}$ are the stress, initial stress and slip velocity, respectively. The superscripts '$ijk$' denote the spatial grids of the fault plane and the temporal steps of the field points, while '$lmn$' denote those of the source points. $C^{ijk,lmn}$ is the kernel that links slip velocity to stress. The detailed expression of $C^{ijk,lmn}$ is presented by Zhang & Chen[63]. For our full-space model, we rewrite Eq. (3) as,

$$\tau^{ijk} = T_e + \sigma_r + \sum_{l,m,n} C^{ijk,lmn} V^{lmn}, \qquad (4)$$

where $T_e = \tau_0 - \sigma_r$. Using Eq. (4) and the slip-weakening frictional law (5), the slip rate and stress at each simulation point at each time step can be obtained.

**Slip-weakening frictional law**. We assume that the simulation area is a strike-slip fault segment embedded in elastic media. The model has a single asperity at the centre with homogeneous frictional properties. Our model is based on the slip-weakening frictional law,

$$T(D) = \begin{cases} \left(1 - \frac{D}{D_c}\right)T_u + \sigma_r, & D < D_c \\ \sigma_r, & D \geq D_c \end{cases}, \qquad (5)$$

where $T(D)$ is the frictional strength, $T_u$ is the breakdown stress drop, $D$ is slip, $D_c$ denotes the critical slip distance, and $\sigma_r$ is the residual stress. By combining Eqs. (4) and (5), we can derive the slip velocity and stress state at every moment on the simulation plane.

We assume LFEs emanate from several single small asperities, so we do not incorporate larger-scale background SSEs and focus on single-asperity rupture behavior. In our model, we use a circle asperity with a homogeneous slip-weakening property surrounded by an unbreakable concentric circle barrier in case run-away ruptures propagate outside our calculation area (Supplementary Fig. S1a). Other geometric asperities can also reproduce the LFE source characteristics, so the choice of circle asperity is only for convenience. To mimic a real earthquake, we set the initial stress in the nucleation zone to be slightly higher than the peak shear stress limit $T_i = 1.001 T_u$. Once rupture is triggered, its subsequent development is controlled by an elastodynamic equation and the slip-weakening frictional law.

**Calculating the phase diagram**. The dynamic rupture of a planar fault depends on the elastic properties of the surrounding media, the parameters of the slip-weakening law and the size of the initial asperity. In our work, we use the same elastic properties throughout the simulation. Thus, we mainly study the influence of the parameters of the slip-weakening law on the rupture scenario.

To compare different scales and frictional parameter ruptures, we use the normalized parameters $\hat{T}_e$ and $\hat{D}_c$,

$$\begin{cases} \hat{T}_e = T_e/T_u \\ \hat{D}_c = D_c/(R_a \frac{T_u}{\mu}) \end{cases}, \qquad (6)$$

where $T_u$ is the breakdown stress drop, $T_e$ is the dynamic stress drop, $D_c$ denotes the critical slip distance, $R_a$ is the effective radius of the nucleation asperity, and $\mu$ is the shear modulus of the media (Fig. 1a). These two parameters together determine the scenario of a rupture simulation.

Here, for calculating the phase diagram, we set the nucleation patch radius as 100 m, $P$ wave velocity as 6.000 km/s, $S$ wave velocity as 3.464 km/s, density as 2.67 g/cm3, background dynamic shear stress drop $T_e$ as 1 MPa, and space step $ds = 3$m for calculation.

The dynamic stress drop $T_e$ varies from 0 to 1 MPa, and the critical slip distance varies from 0 to 3 mm so that the normalized $\hat{T}_e$ and $\hat{D}_c$ vary from 0 to 1. Each pair $(\hat{T}_e, \hat{D}_c)$ represents a simulation scenario, that is, a single point on the phase diagram. By simulating all pairs $(\hat{T}_e, \hat{D}_c)$ from 0 to 1, we obtain the phase diagram and differentiate different rupture types with boundary lines, as shown in Fig. 1.

**Model parameter settings for comparison between LFEs and SSARs**. According to the estimated source parameters of LFEs, the LFE rupture patch diameter is ~200 m. Considering the existence of error in the estimation of the rupture diameter of LFEs, we simulate earthquakes with rupture diameters of 180 m, 220 m, 260 m and 300 m. Since super-shear rupture is clearly too rapid to fit the LFE source characteristics, we focus on sub-Rayleigh and self-arresting ruptures and SSARs as potential rupture models of LFEs. Based on the source model described before, we set four groups of experiments according to their different rupture patch diameters. For each group, we simulate all three kinds of ruptures with different combinations of $\hat{T}_e$ and $\hat{D}_c$ values. All simulated pairs of $(\hat{T}_e, \hat{D}_c)$ are shown in Fig. S 1b and Table S2.

For SSARs, once we set the nucleation zone asperities, the rupture area is determined due to their rupture nature (arrested within the nucleation area). Therefore, we set the diameters of the circle nucleation zone directly to 180 m, 220 m, 260 m and 300 m. For sub-Rayleigh ruptures, we set unbreakable concentric

circle barriers to terminate their propagation. Therefore, we set the unbreakable barrier diameters to 180 m, 220 m, 260 m and 300 m. In regard to SSARs, we cannot predict the patch diameter before rupture arrest by themselves. We conducted a large number of self-arresting rupture experiments and selected events with rupture patch diameters of 180 m, 220 m, 260 m and 300 m. Thus far, we have obtained numerical simulation results of three kinds of ruptures with areas of 180 m, 220 m, 260 m, and 300 m.

**Calculating source parameters of simulated earthquakes**. We calculate the source duration, seismic moment, stress drop, slip and slip rate of each simulated rupture event. Using $T = N \times dt$, where $T$ is the source duration, $N$ is the total time step for one simulation and $dt$ denotes the time step used in the numerical calculation, we derive the source duration. The average stress drop can be expressed as:

$$\Delta\sigma = \frac{1}{M}\sum_i (\sigma_{init} - \sigma)_i, \qquad (7)$$

where $\sigma_{init}$ is the initial shear stress, $\sigma$ is the final shear stress after rupture stops, $M$ represents the total number of points in the simulation where the rupture velocity is not 0, and the integral to $i$ represents the integral to all rupture points in space. In addition, we calculate the average slip of each rupture scenario,

$$D = \frac{1}{M}\sum_i \sum_t \dot{D}_{it}, \qquad (8)$$

where $D$ denotes the average slip and $\dot{D}_{it}$ is the slip rate of every simulated rupture point at moment t, which is calculated at each time step. The average slip rate is divided by the source duration to obtain the average slip rate for each rupture event. According to the definition of the seismic moment $M_0 = \mu AD$, where $\mu$ is the rigidity, $A$ is the slip area, and $D$ is the slip, the seismic moment can be derived. We provide all of the calculated source parameters in the Supplementary Data. S2.

**Estimating source parameters of LFEs on the SAF**. We use borehole seismograms from the High Resolution Seismic Network (HRSN) for estimation analysis. We download all waveform data from the NCEDC, corrected for instrument response. We perform our estimation of LFE source parameters using S waves identified on components DP2 and DP3 filtered at 2–8 Hz. Our method of estimating the moment magnitudes of LFEs mainly refers to Chestler et al.[7]

For far-field S wave displacement, $u^s$, in an elastic whole space observed at position $x$ from a source at $x = 0$:

$$u^s(x,t) = \frac{1}{4\pi\rho\beta^3}(\cos 2\theta \cos \phi\hat{\theta} - \cos\theta \sin \phi\hat{\phi})\frac{1}{r}\dot{M}_0\left(t - \frac{r}{\beta}\right), \qquad (9)$$

where $\beta$ is the shear-wave velocity, $\rho$ is the density, $r$ is the source-receiver distance, and $\dot{M}_0$ is the moment rate. By integrating the moment rate over time, we solve for the seismic moment $M_0$. Based on the CRUST 1.0 model, we also set the transmission coefficient between the velocity and density at the source, $\beta_2 = 4040$m/sand $\rho_2 = 2990$kg/m$^3$, and the velocity and density at the receiver, $\beta_1 = 3500$m/s and $\rho_1 = 2990$kg/m$^3$. Assuming a model with only a source and receiver velocity and near-vertical ray paths, the following equation can be derived:

$$M_0 \approx \frac{4\pi\beta_2^2(\rho_1\beta_1 + \rho_2\beta_2)r}{2Rc}e^{\frac{rf\pi^*}{\beta Q_s}} \text{median}(d_k^2, d_k^3), \qquad (10)$$

where median$(d_k^2, d_k^3)$ is the time integral of the S wave displacement observed at each receiver, $R = 2$ is a free-surface correction for near-vertical incident rays, $c$ denotes the directional term in Eq. (9) over the unit sphere, $f = 2$Hz is the dominant frequency, $\beta$ is the average of $\beta_1$ and $\beta_2$, and $Q_s$ is the S wave quality factor. In this study, we use a quality factor of 200[65,66]. We integrate the displacement of the S wave from 1 s before the S wave arrival to 5 s after the S wave arrival, which is sufficient to obtain a stable estimation result.

According to a previous study of LFEs in Parkfield[8], the slip per LFE in the Parkfield region is ~0.05 mm, and the source duration time scale is ~0.2 s. Using the moment and slip of LFE, we can derive the LFE asperity area and stress drop based on the definition of seismic moment as $M_0 = \mu AD$, where $\mu$ is the rigidity, $A$ is the slip area, and $D$ is the slip. We then define the LFE asperity as having a circle shape with $A = \pi L^2$, where $L$ is the radius of the asperity. In addition, the stress drop is defined as:

$$\Delta\sigma \approx \frac{\mu D}{2L}, \qquad (11)$$

All of the estimated source parameter results of Parkfield LFEs are shown in Supplementary Data S1.

The method of estimating moment magnitude and stress drop depends on the quality factor $Q_s$. Therefore, we also explore the difference between the results under different $Q_s$ values (Supplementary Figs. S7 and S8). With the increase in the $Q_s$ value, the smaller the average moment magnitude of LFEs obtained, the larger the average stress drop. However, the change is small, and the estimated 95% of the LFE source parameters are still within the range that we used to compare with the simulation results. Due to the close horizontal distance between the stations and LFEs, the change in the $Q_s$ value has a limited effect on the estimated magnitude

and stress drop. Thus, a value of 200 for $Q_s$ is relatively reliable in estimating the range of moment magnitude and stress drop of LFEs.

## Data availability

The HRSN borehole seismic data (network code BP) used in this work can be downloaded from the NCEDC (http://www.ncedc.org/). The tremor/LFE catalogue is obtained from Data S1 of Shelly10. The source parameters data of simulated earthquakes generated in this study are provided in the Supplementary Information.

## Code availability

The code for simulation of earthquake ruptures in this work is available at git-hub (https://github.com/wxt5588/RUPTURE_BIEM). The code for calculating source parameters of LFEs is available at git-hub (https://github.com/wxt5588/SeismicMomentCalculation).

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

## Acknowledgements

We thank D. R. Shelly for sharing his tremor data and the valuable comments of reviewers.

## Author contributions

X.W., J.X., Y.L. and X.C. designed the experiments, analyzed the data, and wrote the paper.

## Funding

This work was supported by the National Natural Science Foundation of China (Grants 41790465, 41874054 and U1901602), Key Special Project for Introduced Talents Team of Southern Marine Science and Engineering Guangdong Laboratory (Guangzhou) (GML2019ZD0203), Shenzhen Science and Technology Program (KQTD2017081011 1725321), and Shenzhen Key Laboratory of Deep Offshore Oil and Gas ExplorationTechnology (GrantNo. ZDSYS20190902093007855).

## Competing interests

The authors declare no competing interests.
