## [Peer Review File · Nature Communications]

REVIEWER COMMENTS

Reviewer #1 (Remarks to the Author):

This manuscript presented a physical model to explain various source parameters and behaviors of low-frequency earthquakes. By using a phase diagram of rupture dynamics, they explain LFEs as slow self-arresting ruptures in nature. They also compared the observed source parameters from LFEs at the Parkfield-Cholame section of the San Andreas Fault, and those from their numerical simulations, and showed that the scaling law of LFEs and slow earthquakes are similar to those from regular earthquakes.

Overall, I found the results to be interesting enough to justify a publication in Nature Communication. However, there are many details that need to be changed/updated before it can be accepted. Some of the key references, especially those related to tremor/LFEs at Parkfield, needed to be added. In addition, although the English is understandable, it can still benefit from further improvement/proof reading. I made some suggestions below.

Major comments:

1. My primary concerns are the measurement of source parameters of the LFEs (which are mostly in the supplementary material). The standard method of measuring source parameters these days are the Empirical Green's function (EGF) method, which was used in Thomas et al. (GRL, 2016). However, in this study, they used a relatively simple method (as given in the equations 8 and 9 in page 27) to measure the moment, and then with some assumptions to get the stress drops. How are those measurements compared with the data points measured by Thomas et al. (GRL, 2016)? In addition, there are millions of LFEs in the Shelly (JGR, 2017) catalog. I do not think that the authors did millions of measurements, right? If so, what criteria did they use to select the LFEs in their analysis? Finally, they used a relatively simple velocity model (Crust 1,0) and a quality factor of 700. There are more complicated 1D or 3D velocity model in this region, and the quality factor for seismic stations near faults might be much smaller. How would the choice of these parameters affect the results? This needs to be further explored/discussed in the paper.
2. Equation 9 in page 27: might be better to change d^N_k and d^E_k to D^2_k and D^3_k since the two horizontal components are not aligned in the N-S and E-W direction. In addition, the authors filtered the data only in the 2-8 Hz filter band. What is the justification for this range? Would the inclusion of high-frequency signal help in determining the corner frequency/moment?
3. Pages 7 and 8, lines 133-134: please refer to the supp. Material here so that the readers know where to find additional information on how those measurements are made. Please specify how many measurements are made for the LFEs here. In addition, the authors only used the data from 2002 to 2010, while the catalog lasted much longer. Why not use the data after 2010?

Figures:

4. Figure 1b shows three types of ruptures with different parameters. However, in the phase diagram in panel c, there are four types. Although it is not the focus of this study, I wonder if it would make sense to include the last type "Super-shear Rupture" in panel b as well (for completeness).
5. Figure 2a: please add the fault map to show the San Andreas Fault as the background. In addition, it would be useful to add an inset to mark the study region in a larger map of CA or US (similar to the supp. Figure S5). Figure 2b and 2c: it is hard to see individual measurements. I wonder if it would make sense to add a separate panel to the right to show the histogram. There are some abnormally high or low values in both plots. What is the cause of these measurements?
6. Figure 3: are there any white stars near the right side of each panel? In figure caption, change "pentagrams" to "stars".

Additional minor comments:

Page 2, line 37-38: please try to cite a few references that have measured the stress-drop of LFEs (e.g., Thomas et al., GRL, 2016, which was cited later).

Page 3, line 42-43: demonstrate their similarity?

Page 3, line 55: change "Parkfield" to "the Parkfield-Cholame section of the San Andreas Fault (SAF), where tremor and LFEs have been recently observed (Nadeau and Guilhem, 2019; Shelly, 2017).".

Page 3, line 57: add "Shelly (2010a, 2010b)" after the "characteristics of multiple LFEs".

Page 3, line 59-60: change the sentence to something like "connects with what controls ..." or "the underlying mechanism is the same for regular earthquakes".

Page 4, line 60: change 'slow' to slow (no need to quote it since it has been mentioned before without quotation).

Page 4, line 76: perhaps start a new paragraph here.

Page 4, line 82: change "And the output" to "The output". Can you specify what are the output (rupture process)?

Page 4, line 86: change "100m, a spatial" to "100 m and a spatial". Can you justify the choice of 100m radius?

Page 6, lines 108-109: perhaps change the last sentence to something like this to make it clear: "Evidence of self-arresting events are observed for repeating earthquakes along the Parkfield section of the SAF14".

Page 7, Line 115-116: change "More extraordinarily" to "In particular" or something similar.

Page 7, line 122: change "Once" to "When" or "If".

Page 7, line 127: change "shows that" to "confirms this inference" or "confirms this observation".

Page 7, line 129: remove "generally" or change "the similarity" to "the general similarity".

Page 8, lines 135-136: The last sentence "And we show..." can be removed by putting the reference to Fig. 2 to the previous sentence like " from 2002 to 2010 (Fig. 2)".

Page 10, line 155: the sentence ends with "to account for" does not seem to be properly finished. Please correct.

Page 10, lines 165-166: change "a long source duration earthquake" to "an earthquake with long-source duration".

Page 11, line 176: change "is required to accumulate to trigger" to "is needed to trigger".

Page 11, line 185: it might useful to add the reference of Peng and Gomberg (NGEO, 2010), who noted similar scaling laws like in Ide et al. (Nature, 2007).

Peng, Z. and J. Gomberg (2010), An integrated perspective of the continuum between earthquakes and slow-slip phenomena, *Nature Geosci.*, 3, 599–607, doi:10.1038/ngeo940.

Page 11, line 188: change "slow self-arresting rupture" to "SSAR".

Page 12, line 194: change "remain" to "keep".

Page 13, lines 199-200: change the sentence to "law, which is similar to regular earthquakes and is consistent with the newest studies19-22".

Page 13, line 204: add some references for the moment magnitudes of VLFs.

Page 13, lines 205-206: remove the words "The fact is that,".

Page 14, line 217: change "which are SSARs", to "which are equivalent to the SSARs in our numerical simulations".

Page 14, line 218: change "stopping" to "rupture arrest".

Page 14, line 226: change "nearby" to "external".

Page 14, line 227: change "Love wave" to "body/surface waves from regional and teleseismic earthquakes".

Page 14, line 227: please add a few references on the tidal triggering and remotely triggered tremor at the Parkfield-Cholame section of the SAF.

Thomas, A.M., R. M. Nadeau, and R. Burgmann (2009) Tremor-tide correlations and near-lithostatic pore pressure on the deep San Andreas fault. *Nature*. doi:10.1038/nature08654

Peng, Z., J. E. Vidale, K. C. Creager, J. L. Rubinstein, J. Gomberg, and P. Bodin (2008), Strong tremor near Parkfield, CA excited by the 2002 Denali Fault earthquake, *Geophys. Res. Lett.*, 35, L23305, doi:10.1029/2008GL036080.

Peng, Z., J. E. Vidale, A. Wech, R. M. Nadeau and K. C. Creager (2009), Remote triggering of tremor along the San Andreas fault in central California, *J. Geophys. Res.*, 114, B00A06, doi:10.1029/2008JB006049.

Shelly, D. R., Z. Peng, D. P. Hill and C. Aiken (2011), Triggered creep as a possible mechanism for delayed dynamic triggering of tremor and earthquakes, *Nature Geosci.*, 4, 384–388, doi: 10.1038/ngeo1141.

Peng, Z., D. R. Shelly, and W. L. Ellsworth (2015), Delay dynamic triggering of deep tremor along the Parkfield-Cholame section of the San Andreas Fault following the 2014 M6.0 South Napa earthquake, *Geophys. Res. Lett.*, 42, 7916–7922, doi:10.1002/2015GL065277.

Page 15, line 246: change "increasing the possibility of LFEs are SSARs in nature" to "furthering confirming our inference that LFEs are SSARs in nature".

Page 15, line 251: change "And tremor" to "Hence, tremor".

Page 16, line 262: change "which characterized" to "which is characterized".

Page 16, line 264: change "distance provide perfect explanation for the swarm-like LFEs" to "distance, provide a natural explanation for the swarm-like behavior of LFEs".

In Supplementary Material:

Page 22, line 381: please add some references after the first sentence (rupture problems).

Page 22, lines 381-382: change "And BIEM (Boundary Integral Equation Method)" to "Among them, the Boundary Integral Equation Method (BIEM)".

Page 25, lines 449: change "characteristics. We" to "characteristics, we".

Page 25, line 458: remove "since".

Page 27, line 488: add "of LFE source parameters" after "estimation".

By Zhigang Peng (Georgia Tech)

Reviewer #2 (Remarks to the Author):

The paper presents a provocative interpretation of low frequency earthquakes based on the results of a numerical analysis of the rupture from which a phase diagram is proposed. In this diagram, the authors define a class of slow self-arresting ruptures and suggest that they correspond to SSEs in nature. This is an interesting study that potentially deserves publication, as it provides elements of interpretation in a discussion that is very active today. The authors limit themselves to a slip model with slip dependent friction. Of course, it is a simple physical model, and therefore it can be criticized in view of the complexity of nature, but it is the interest of this kind of approach to propose the simplest arguments and to show that they are compatible with the main characteristics of the observations.

Nevertheless, I have some questions and remarks that I think should be considered by the authors.

1) The question of the evolution towards instability of a surface with slip-dependent weakening has been treated in theoretical papers which describe the relations between initial perturbation, weakening rate and evolution time (for example Campillo, M. and I. Ionescu (1997), Initiation of antiplane shear instability under slip dependent friction, *Journal of Geophysical Research*, 102, 20363-20371.). Moreover, the evolution on a surface of finite size and the inhibition of the instability has been precisely studied (for example: Dascalu, C., I.R. Ionescu and M. Campillo (2000), Fault Finiteness and Initiation of Dynamic Shear Instability, *Earth and Planetary Science Letters*, 177, 163-1766., Uenishi and Rice (2003), Universal nucleation length for slip-weakening rupture instability under nonuniform fault loading *Journal of Geophysical Research* <https://doi.org/10.1029/2001JB001681>). I believe that these previous results are relevant for the understanding of the existence of the class of SSARs described in the paper.

2) In this regime, (and in general in the nucleation zone) there is no propagative stress concentration and therefore the notion of rupture velocity is not defined. This should be clarified because the distance-time passage used in the discussion on scaling seems to me very arbitrary.

3) The model is built with an initial forcing in a central zone and this point should be discussed precisely because it controls the duration, and potentially the final evolution (see point 1) for SSARs, that is in absence of unstable growth. I wonder what is the role of the initial conditions on the spectrum that is represented in Figure S4. Clarifying this issue seems very important in relation to the results showing a good quantitative correspondence between the model predictions and the observations. The authors should show that this success is not depending on an ad-hoc choice of the initial triggering.

Reviewer #3 (Remarks to the Author):

General Comments

Low-frequency earthquakes (LFEs) are small events (magnitudes roughly between about 0.7 and 2.2) that occur during episodes of non-volcanic tremor. They are characterized by unusually long source durations, small slip rates, and small stress drops. Their source mechanics remain a mystery.

In this paper Wei et al. show that a standard 3D elastic fault model, with a slip-weakening friction law, can generate an LFE having these observed characteristics. Whether the model generates an LFE or a standard earthquake depends on the stress-state and friction parameters. The parameter space they explore is defined by two dimensionless parameters:

$$T^*_e = T_e / T_u \text{ and } D^*_c = D_c / ((R_a T_u / \mu))$$

Where T_u is the breakdown stress, T_e is the dynamic stress drop, D_c is the critical slip distance in the slip-weakening friction law, and μ is the shear modulus of the medium.

In this (D_c^*, T_e^*) space LFE type ruptures occur at large values of D_c^* ($D_c^* > 0.85$) and for nearly the entire range of T_e^* , ($0 < T_e^* < 1$).

Beyond the general framework outlined above, I found this paper very difficult to follow. I do not question that they did the numerical simulations correctly, but the physical significance of the model parameters was not obvious to me. Granted, I am not an expert on seismic source modeling, but I am close enough to the field that if I don't understand, I suspect that the broader audience of Nature readers may have similar problems.

To be specific, the key element of the physics here is that rupture propagation in an LFE is arrested during the nucleation process. It seems important therefore that this nucleation process should be described in detail. For example, how do the authors nucleate a rupture in their model? Do they introduce a heterogeneity at the center of the nucleation patch? Once nucleated, what physically determines whether the rupture will die before it reaches the critical radius R_c for run-away propagation thereby producing an LFE?

I do not find it surprising that LFE production occurs at the largest values of the scaled critical radius ($D_c^* > 0.85$). Generally speaking, rupture propagation is a competition between decreasing friction (slip-weakening in this case) and decreasing elastic driving stress that accompanies displacement on the fault. If the fault stiffness, which depends on the radius of the slipping patch, decreases faster than the resisting friction, then slip will be arrested. (Dieterich (1986) demonstrated that such considerations lead to a minimum critical radius R_c required for rupture propagation, which depends on D_c . For larger values of D_c , the friction resistance to slip decreases more slowly with displacement, and the critical radius for run-away propagation is larger. and for the same elastic parameters, it is more likely that the rupture will be arrested before it reaches R_c .

Dieterich, J. H. (1986), A model for the nucleation of earthquake slip. In: S. Das, J. Boatwright, and C.H. Scholz (Editors), Earthquake Source Mechanics. Am. Geophys. Union, M. Ewing Vol. 6, Geophys. Monogr., 37, 37-47.

What is missing in this paper is some sense of whether the friction parameters that produce an LFE make physical sense. If the LFEs arrest at a radius near 200 meters as claimed here, what is the corresponding value of D_c ? How does this compare with values of D_c found for normal earthquakes? Why is D_c larger at the base of the seismogenic zone where LFEs are observed than at shallower depths where normal earthquakes occur?

Finally, I am not convinced by the statement (line 186) "Accumulating new evidences show that slow earthquakes follow a cubic moment-duration scaling law similar to regular earthquakes". This is a prediction of their model which they would like to be true. However, as far as I can tell, the jury is still out on moment duration scaling as evidenced by the following (2020) JGR paper

Farge, Shapiro, and Frank, 2020. Moment-Duration Scaling of Low-Frequency Earthquakes in Guerrero, Mexico. JGR Solid Earth, 124 (8).
<https://doi.org/10.1029/2019JB019099>

"We find characteristic values of $M_0 \sim 3 \times 10^{12}$ N.m ($M_w \sim 2.3$) and $f_c \sim 3.0$ Hz with the corner frequency very weakly dependent on the seismic moment. This moment-duration scaling observed for Mexican LFEs is similar to one previously reported in Cascadia and is very different from the established one for regular earthquakes. This suggests that they could be generated by sources of nearly constant size with strongly varying intensities. LFEs do not exhibit the self-similarity characteristic of regular earthquakes, suggesting that the physical mechanisms at their origin could be intrinsically different."

The Cascadia papers that find LFE duration is approximately independent of its magnitude are (Bostock et al., 2015; Thomas et al. 2016).

Bostock, M.G., Thomas, A. M., Savard G., Chuang L., & Rubin, A. M. (2015). Magnitudes and moment-duration scaling of low-frequency earthquakes beneath southern Vancouver Island, *J. Geophys. Res. Solid Earth*, 120, <https://doi.org/10.1002/2015JB012195>

Thomas, A. M., Beroza, G. C., and Shelly, D. R. (2016). Constraints on the source parameters of low-frequency earthquakes on the San Andreas Fault, *Geophys. Res. Lett.*, 43, 1464– 1471, [doi:10.1002/2015GL067173](https://doi.org/10.1002/2015GL067173).

Specific line-by-line comments

15. and 33. Is it established that volcanic tremor consists entirely of LFEs?

42. Waveform correlations between LFEs and regular earthquakes suggest that they share a similar source process. Is this true?

52. how is the nucleation zone defined?

116. How do you set the patch in the simulation to initiate the earthquake.

137. Are these estimates from your model or the data?

139. where does the number 240 m for the rupture patch diameters come from?

Summary

In summary the major claim in this paper is that it is possible to produce LFE events using a standard earthquake model under special conditions of stress and friction. I am not convinced that this demonstration is of sufficient general interest to warrant publication in *Nature*, especially in view of their limited physical interpretation of the parameters in their model and the controversial nature of many of the observations offered in support. More suitable venues might be the *Journal of the Seismological Society of America* or the *Journal of Geophysical Research* which are aimed at a more specialized audience that could better assess the importance of this paper in the context of the extensive research on Low Frequency Earthquakes.

Reviewers' comments

Reviewer #1 (Remarks to the Author):

This manuscript presented a physical model to explain various source parameters and behaviors of low-frequency earthquakes. By using a phase diagram of rupture dynamics, they explain LFEs as slow self-arresting ruptures in nature. They also compared the observed source parameters from LFEs at the Parkfield-Cholame section of the San Andreas Fault, and those from their numerical simulations, and showed that the scaling law of LFEs and slow earthquakes are similar to those from regular earthquakes.

Overall, I found the results to be interesting enough to justify a publication in Nature Communication. However, there are many details that need to be changed/updated before it can be accepted. Some of the key references, especially those related to tremor/LFEs at Parkfield, needed to be added. In addition, although the English is understandable, it can still benefit from further improvement/proof reading. I made some suggestions below.

Major comments:

1. My primary concerns are the measurement of source parameters of the LFEs (which are mostly in the supplementary material). The standard method of measuring source parameters these days are the Empirical Green's function (EGF) method, which was used in Thomas et al. (GRL, 2016). However, in this study, they used a relatively simple method (as given in the equations 8 and 9 in page 27) to measure the moment, and then with some assumptions to get the stress drops. How are those measurements compared with the data points measured by Thomas et al. (GRL, 2016)? In addition, there are millions of LFEs in the Shelly (JGR, 2017) catalog. I do not think that the authors did millions of measurements, right? If so, what criteria did they use to select the LFEs in their analysis? Finally, they used a relatively simple velocity model (Crust 1,0) and a quality factor of 700. There are more complicated 1D or 3D velocity model in this region, and the quality factor for seismic stations near faults might be much smaller. How would the choice of these parameters affect the results? This needs to be further explored/discussed in the paper.
2. Equation 9 in page 27: might be better to change d^N_k and d^E_k to D^2_k and d^3_k since the two horizontal components are not aligned in the N-S and E-W direction. In addition, the authors filtered the data only in the 2-8 Hz filter band. What is the justification for this range? Would the inclusion of high-frequency signal help in determining the corner frequency/moment?
3. Pages 7 and 8, lines 133-134: please refer to the supp. Material here so that the readers know where to find additional information on how those measurements are made. Please specify how many measurements are made for the LFEs here. In addition, the authors only used the data from 2002 to 2010, while the catalog lasted much longer. Why not use the data after 2010?

Figures:

4. Figure 1b shows three types of ruptures with different parameters. However, in the phase diagram in panel c, there are four types. Although it is not the focus of this study, I wonder if it would make sense to include the last type “Super-shear Rupture” in panel b as well (for completeness).

5. Figure 2a: please add the fault map to show the San Andreas Fault as the background. In addition, it would be useful to add an inset to mark the study region in a larger map of CA or US (similar to the supp. Figure S5). Figure 2b and 2c: it is hard to see individual measurements. I wonder if it would make sense to add a separate panel to the right to show the histogram. There are some abnormally high or low values in both plots. What is the cause of these measurements?

6. Figure 3: are there any white stars near the right side of each panel? In figure caption, change “pentagrams” to “stars”.

Additional minor comments:

Page 2, line 37-38: please try to cite a few references that have measured the stress-drop of LFEs (e.g., Thomas et al., GRL, 2016, which was cited later).

Page 3, line 42-43: demonstrate their similarity?

Page 3, line 55: change “Parkfield” to “the Parkfield-Cholame section of the San Andreas Fault (SAF), where tremor and LFEs have been recently observed (Nadeau and Guilhem, 2019; Shelly, 2017).”.

Page 3, line 57: add “Shelly (2010a, 2010b)” after the “characteristics of multiple LFEs”.

Page 3, line 59-60: change the sentence to something like “connects with what controls ...” or “the underlying mechanism is the same for regular earthquakes”.

Page 4, line 60: change ‘slow’ to slow (no need to quote it since it has been mentioned before without quotation).

Page 4, line 76: perhaps start a new paragraph here.

Page 4, line 82: change “And the output” to “The output”. Can you specify what are the output (rupture process)?

Page 4, line 86: change “100m, a spatial” to “100 m and a spatial”. Can you justify the choice of 100m radius?

Page 6, lines 108-109: perhaps change the last sentence to something like this to make it clear: "Evidence of self-arresting events are observed for repeating earthquakes along the Parkfield section of the SAF14".

Page 7, Line 115-116: change "More extraordinarily" to "In particular" or something similar.

Page 7, line 122: change "Once" to "When" or "If".

Page 7, line 127: change "shows that" to "confirms this inference" or "confirms this observation".

Page 7, line 129: remove "generally" or change "the similarity" to "the general similarity".

Page 8, lines 135-136: The last sentence "And we show..." can be removed by putting the reference to Fig. 2 to the previous sentence like "from 2002 to 2010 (Fig. 2)".

Page 10, line 155: the sentence ends with "to account for" does not seem to be properly finished. Please correct.

Page 10, lines 165-166: change "a long source duration earthquake" to "an earthquake with long-source duration".

Page 11, line 176: change "is required to accumulate to trigger" to "is needed to trigger".

Page 11, line 185: it might useful to add the reference of Peng and Gomberg (NGEO, 2010), who noted similar scaling laws like in Ide et al. (Nature, 2007).

Peng, Z. and J. Gomberg (2010), An integrated perspective of the continuum between earthquakes and slow-slip phenomena, Nature Geosci., 3, 599–607, doi:10.1038/ngeo940.

Page 11, line 188: change "slow self-arresting rupture" to "SSAR".

Page 12, line 194: change "remain" to "keep".

Page 13, lines 199-200: change the sentence to "law, which is similar to regular earthquakes and is consistent with the newest studies19-22".

Page 13, line 204: add some references for the moment magnitudes of VLFs.

Page 13, lines 205-206: remove the words "The fact is that,".

Page 14, line 217: change “which are SSARs”, to “which are equivalent to the SSARs in our numerical simulations”.

Page 14, line 218: change “stopping” to “rupture arrest”.

Page 14, line 226: change “nearby” to “external”.

Page 14, line 227: change “Love wave” to “body/surface waves from regional and teleseismic earthquakes”.

Page 14, line 227: please add a few references on the tidal triggering and remotely triggered tremor at the Parkfield-Cholame section of the SAF.

Thomas, A.M., R. M. Nadeau, and R. Burgmann (2009) Tremor-tide correlations and near-lithostatic pore pressure on the deep San Andreas fault. *Nature*. doi:10.1038/nature08654

Peng, Z., J. E. Vidale, K. C. Creager, J. L. Rubinstein, J. Gomberg, and P. Bodin (2008), Strong tremor near Parkfield, CA excited by the 2002 Denali Fault earthquake, *Geophys. Res. Lett.*, 35, L23305, doi:10.1029/2008GL036080.

Peng, Z., J. E. Vidale, A. Wech, R. M. Nadeau and K. C. Creager (2009), Remote triggering of tremor along the San Andreas fault in central California, *J. Geophys. Res.*, 114, B00A06, doi:10.1029/2008JB006049.

Shelly, D. R., Z. Peng, D. P. Hill and C. Aiken (2011), Triggered creep as a possible mechanism for delayed dynamic triggering of tremor and earthquakes, *Nature Geosci.*, 4, 384–388, doi: 10.1038/ngeo1141.

Peng, Z., D. R. Shelly, and W. L. Ellsworth (2015), Delay dynamic triggering of deep tremor along the Parkfield-Cholame section of the San Andreas Fault following the 2014 M6.0 South Napa earthquake, *Geophys. Res. Lett.*, 42, 7916-7922, doi:10.1002/2015GL065277.

Page 15, line 246: change “increasing the possibility of LFEs are SSARs in nature” to “furthering confirming our inference that LFEs are SSARs in nature”.

Page 15, line 251: change “And tremor” to “Hence, tremor”.

Page 16, line 262: change “which characterized” to “which is characterized”.

Page 16, line 264: change “distance provide perfect explanation for the swarm-like LFEs” to “distance, provide a natural explanation for the swarm-like behavior of LFEs”.

In Supplementary Material:

Page 22, line 381: please add some references after the first sentence (rupture problems).

Page 22, lines 381-382: change “And BIEM (Boundary Integral Equation Method)” to “Among them, the Boundary Integral Equation Method (BIEM)”.

Page 25, lines 449: change “characteristics. We” to “characteristics, we”.

Page 25, line 458: remove “since”.

Page 27, line 488: add “of LFE source parameters” after “estimation”.

By Zhigang Peng (Georgia Tech)

Reviewer #2 (Remarks to the Author):

The paper presents a provocative interpretation of low frequency earthquakes based on the results of a numerical analysis of the rupture from which a phase diagram is proposed. In this diagram, the authors define a class of slow self-arresting ruptures and suggest that they correspond to SSEs in nature. This is an interesting study that potentially deserves publication, as it provides elements of interpretation in a discussion that is very active today. The authors limit themselves to a slip model with slip dependent friction. Of course, it is a simple physical model, and therefore it can be criticized in view of the complexity of nature, but it is the interest of this kind of approach to propose the simplest arguments and to show that they are compatible with the main characteristics of the observations.

Nevertheless, I have some questions and remarks that I think should be considered by the authors.

1) The question of the evolution towards instability of a surface with slip-dependent weakening has been treated in theoretical papers which describe the relations between initial perturbation, weakening rate and evolution time (for example Campillo, M. and I. Ionescu (1997), Initiation of antiplane shear instability under slip dependent friction, *Journal of Geophysical Research*, 102, 20363-20371.). Moreover, the evolution on a surface of finite size and the inhibition of the instability has been precisely studied

(for example: Dascalu, C., I.R. Ionescu and M. Campillo (2000), Fault Finiteness and Initiation of Dynamic Shear Instability, *Earth and Planetary Science Letters*, 177, 163-176., Uenishi and Rice (2003), Universal nucleation length for slip-weakening rupture instability under nonuniform fault loading *Journal of Geophysical Research* <https://doi.org/10.1029/2001JB001681>). I believe that these previous results are relevant for the understanding of the existence of the class of SSARs described in the paper.

2) In this regime, (and in general in the nucleation zone) there is no propagative stress concentration and therefore the notion of rupture velocity is not defined. This should be clarified because the distance-time passage used in the discussion on scaling seems to me very arbitrary.

3) The model is built with an initial forcing in a central zone and this point should be discussed precisely because it controls the duration, and potentially the final evolution (see point 1) for SSARs, that is in absence of unstable growth. I wonder what is the role of the initial conditions on the spectrum that is represented in Figure S4. Clarifying this issue seems very important in relation to the results showing a good quantitative correspondence between the model predictions and the observations. The authors should show that this success is not depending on an ad-hoc choice of the initial triggering.

Reviewer #3 (Remarks to the Author):

General Comments

Low-frequency earthquakes (LFEs) are small events (magnitudes roughly between about 0.7 and 2.2) that occur during episodes of non-volcanic tremor. They are characterized by unusually long source durations, small slip rates, and small stress drops. Their source mechanics remain a mystery.

In this paper Wei et al. show that a standard 3D elastic fault model, with a slip-weakening friction law, can generate an LFE having these observed characteristics. Whether the model generates an LFE or a standard earthquake depends on the stress-state and friction parameters. The parameter space they explore is defined by two dimensionless parameters:

$$T_e = T_e / T_u \text{ and } D_c = D_c / ((R_a T_u / \mu))$$

Where T_u is the breakdown stress, T_e is the dynamic stress drop, D_c is the critical slip distance in the slip-weakening friction law, and μ is the shear modulus of the medium.

In this (D_c , T_e) space LFE type ruptures occur at large values of D_c ($D_c > 0.85$) and for nearly the entire range of T_e , ($0 < T_e < 1$).

Beyond the general framework outlined above, I found this paper very difficult to

follow. I do not question that they did the numerical simulations correctly, but the physical significance of the model parameters was not obvious to me. Granted, I am not an expert on seismic source modeling, but I am close enough to the field that if I don't understand, I suspect that the broader audience of Nature readers may have similar problems.

To be specific, the key element of the physics here is that rupture propagation in an LFE is arrested during the nucleation process. It seems important therefore that this nucleation process should be described in detail. For example, how do the authors nucleate a rupture in their model? Do they introduce a heterogeneity at the center of the nucleation patch? Once nucleated, what physically determines whether the rupture will die before it reaches the critical radius R_c for run-away propagation thereby producing an LFE?

I do not find it surprising that LFE production occurs at the largest values of the scaled critical radius ($D_c > 0.85$). Generally speaking, rupture propagation is a competition between decreasing friction (slip-weakening in this case) and decreasing elastic driving stress that accompanies displacement on the fault. If the fault stiffness, which depends on the radius of the slipping patch, decreases faster than the resisting friction, then slip will be arrested. (Dieterich (1986) demonstrated that such considerations lead to a minimum critical radius R_c required for rupture propagation, which depends on D_c . For larger values of D_c , the friction resistance to slip decreases more slowly with displacement, and the critical radius for run-away propagation is larger. and for the same elastic parameters, it is more likely that the rupture will be arrested before it reaches R_c .

Dieterich, J. H. (1986), A model for the nucleation of earthquake slip. In: S. Das, J. Boatwright, and C.H. Scholz (Editors), Earthquake Source Mechanics. Am. Geophys. Union, M. Ewing Vol. 6, Geophys. Monogr., 37, 37-47.

What is missing in this paper is some sense of whether the friction parameters that produce an LFE make physical sense. If the LFEs arrest at a radius near 200 meters as claimed here, what is the corresponding value of D_c ? How does this compare with values of D_c found for normal earthquakes? Why is D_c larger at the base of the seismogenic zone where LFEs are observed than at shallower depths where normal earthquakes occur?

Finally, I am not convinced by the statement (line 186) "Accumulating new evidences show that slow earthquakes follow a cubic moment-duration scaling law similar to regular earthquakes". This is a prediction of their model which they would like to be true. However, as far as I can tell, the jury is still out on moment duration scaling as evidenced by the following (2020) JGR paper

Farge, Shapiro, and Frank, 2020. Moment-Duration Scaling of Low-Frequency

Earthquakes in Guerrero, Mexico. JGR Solid Earth, 124 (8).

<https://doi.org/10.1029/2019JB019099>

“We find characteristic values of $M_0 \sim 3 \times 10^{12}$ N.m ($M_w \sim 2.3$) and $f_c \sim 3.0$ Hz with the corner frequency very weakly dependent on the seismic moment. This moment-duration scaling observed for Mexican LFEs is similar to one previously reported in Cascadia and is very different from the established one for regular earthquakes. This suggests that they could be generated by sources of nearly constant size with strongly varying intensities. LFEs do not exhibit the self-similarity characteristic of regular earthquakes, suggesting that the physical mechanisms at their origin could be intrinsically different.”

The Cascadia papers that find LFE duration is approximately independent of its magnitude are (Bostock et al., 2015; Thomas et al. 2016).

Bostock, M.G., Thomas, A. M., Savard G., Chuang L., & Rubin, A. M. (2015). Magnitudes and moment-duration scaling of low-frequency earthquakes beneath southern Vancouver Island, J. Geophys. Res. Solid Earth, 120, <https://doi.org/10.1002/2015JB012195>

Thomas, A. M., Beroza, G. C., and Shelly, D. R. (2016). Constraints on the source parameters of low-frequency earthquakes on the San Andreas Fault, Geophys. Res. Lett., 43, 1464– 1471, doi:10.1002/2015GL067173.

Specific line-by-line comments

15. and 33. Is it established that volcanic tremor consists entirely of LFEs?

42. Waveform correlations between LFEs and regular earthquakes suggest that they share a similar source process. Is this true?

52. how is the nucleation zone defined?

116. How do you set the patch in the simulation to initiate the earthquake.

137. Are these estimates from your model or the data?

139. where does the number 240 m for the rupture patch diameters come from?

Summary

In summary the major claim in this paper is that it is possible to produce LFE events using a standard earthquake model under special conditions of stress and friction. I

am not convinced that this demonstration is of sufficient general interest to warrant publication in Nature, especially in view of their limited physical interpretation of the parameters in their model and the controversial nature of many of the observations offered in support. More suitable venues might be the Journal of the Seismological Society of America or the Journal of Geophysical Research which are aimed at a more specialized audience that could better assess the importance of this paper in the context of the extensive research on Low Frequency Earthquakes.

We thank all reviewers for the constructive feedback. We appreciate the thoughtful and positive comments, which have certainly helped improve the presentation and quality of our paper. We have updated our paper according to the suggestions and performed more experiments as requested by the reviewers.

In this revised manuscript, we carefully consider the comments made by all reviewers and have made the following major changes:

1. We used more data to measure LFEs' source parameters. We used a more accurate quality factor during estimation and discussed the influence of different quality factors and frequency filter bands on our measurement results.
2. We added detailed descriptions on our nucleation method.
3. We added a different nucleation triggering method in Supplementary Text to show that our result is not depending on an ad-hoc choice of the initial triggering.
4. We added a model of LFEs and regular earthquakes to illustrate the specific physical meanings of the frictional parameters.
5. We modified and re-simulated SSARs in the scaling discussion section to reduce ambiguity.
6. We added more experiments to illustrate the observed phenomenon of LFEs mentioned by Reviewer #3.
7. We proofread the manuscript and revised the figures.

The answers to the reviewers are shown in blue.

Modifications of the manuscript in orange

Reviewer #1 (Remarks to the Author):

This manuscript presented a physical model to explain various source parameters and behaviors of low-frequency earthquakes. By using a phase diagram of rupture dynamics, they explain LFEs as slow self-arresting ruptures in nature. They also compared the observed source parameters from LFEs at the Parkfield-Cholame section of the San Andreas Fault, and those from their numerical simulations, and showed that the scaling law of LFEs and slow earthquakes are similar to those from regular earthquakes.

Overall, I found the results to be interesting enough to justify a publication in Nature Communication. However, there are many details that need to be changed/updated before it can be accepted. Some of the key references, especially those related to tremor/LFEs at Parkfield, needed to be added. In addition, although the English is understandable, it can still benefit from further improvement/proof reading. I made some suggestions below.

Major comments:

1. My primary concerns are the measurement of source parameters of the LFEs (which are mostly in the supplementary material). The standard method of measuring source parameters these days are the Empirical Green's function(EGF) method, which was used in Thomas et al. (GRL, 2016). However, in this study, they used a relatively simple method (as given in the equations 8 and 9 in page 27) to measure the moment, and then with some assumptions to get the stress drops. How are those measurements compared with the data points measured by Thomas et al. (GRL, 2016)?

This is a very good question, and we thank the reviewer for this valuable question. First, we wish to clarify that our main purpose is to obtain the magnitude and stress drop of a single low-frequency earthquake so that the magnitude and stress drop range of all LFEs in the Parkfield area can be obtained. Then, this range is used to compare with the simulation results.

For the duration range of LFEs in Parkfield, we rely directly on the results of Thomas et al. (GRL, 2016)¹ because they use the same data as ours. For the magnitude and stress drop, we use the method proposed by Chestler et al. (JGR, 2017)² as presented in our article. As claimed by Chestler et al.², for this method, *"We believe it is a good enough approximation given other sources of error during the inversion process (e.g., uncertainty of attenuation, the exact density/velocity structure of the subsurface, the focal mechanism, and the signal-to-noise ratio for these tiny events)."* The reasons for not using the empirical Green function (EGF) method to measure the magnitude of LFEs are as follows:

- 1) Too few LFEs that can be used to measure magnitude (using the EGF)
Although there are millions of LFEs in the Shelly (JGR, 2017)³ catalogue, the signal-to-noise ratios are all very low. This causes the correlation coefficient of a single LFE and the EGF to be too low, so the EGF method cannot be used. Thus, to improve the correlation ratios to use the EGF method, Thomas et al.¹ superimposed LFEs of the same family, which means that all LFEs in a family are regarded as one low-frequency earthquake in the measurement. The superimposition process greatly reduces the number of LFEs of measurement. For example, Thomas et al.¹ actually only measured the average source duration of two superimposed LFE families (families 37102 and 37140). However, our goal is to measure every single LFE in all families in Shelly's catalogue to obtain the magnitude and stress drop range of all LFEs in Parkfield. We are more concerned about how large a single LFE can be rather than the average magnitude of an LFE family.
- 2) Too few EGFs
Due to the deep location of LFEs, most LFEs do not have a suitable EGF according to the EGF selection criteria. Thomas et al. selected all relocated earthquakes^{4,5} with hypocentres at distances of less than 3 km from the

respective LFE family and with catalogue magnitudes of 1.4 or below. Based on the same criteria, we also used the same criteria to search all suitable EGFs for all families. We used the same latest relocated earthquake catalogue measured by Waldhauser and Schaff⁵ as Thomas et al.¹ and followed the same requirements; that is, the distance to LFEs is less than 3 km, and the magnitude is equal to or less than 1.4 to search EGFs. We found that for our 6585 LFEs, 6159 LFEs have 0 EGFs, 246 LFEs have 1 EGF, 97 LFEs have 2 EGFs, and 83 LFEs have 15 EGFs. Through further analysis, it can be found that 83 LFEs with 15 EGFs are all in the 37102 family. The 60 LFEs with 2 EGFs include the 37140 and 21462 families (Thomas et al.¹ showed that 37140 family earthquakes have 10 EGFs under this standard, which may be because the relocation database has been updated and the distance calculation method is different. If we relax the standard from 3.2 km, then the 37140 family will also have 10 EGFs). In addition, 246 LFEs with 1 EGF belong to 6 families. According to our results, the 37102 and 37140 families mentioned in Thomas et al.¹'s paper are the most suitable families for the EGF method. *However, there are too many LFEs (most LFEs) without suitable EGFs, which is also the reason that it is essentially difficult to estimate the source parameters of each LFE with the EGF method.* For LFE families with multiple EGFs, using the EGF method means stacking all LFEs in those families instead of obtaining the source parameters of a single LFE, which is contrary to our original intention. If we use the EGF method, we can measure the source parameters of only 9 stacked LFEs at most.

3) Low cross-correlation coefficient

The traditional EGF method requires a high correlation coefficient between EGFs and seismic events, but the correlation coefficient between LFEs and EGFs is very small. Although using the stacked EGF and stacked LFEs, the correlation results are still poor. As Thomas et al.¹ wrote, for stacked LFEs, *"While we did experiment with this technique, it did not produce reliable estimates of duration due to the combination of low SNR in many of the eGfs and the band-limited nature of LFEs."* Therefore, they adopted an innovative method of reverse fitting to measure the average source duration of LFEs. However, in theory, we do not think that this method will significantly improve the measurement quality of seismic magnitude or stress drop.

Based on the above three points, especially the second point, we choose to use Chestler et al.²'s method rather than the EGF method to measure the magnitude and stress drop of LFEs. Here, we use the EGF method to measure the magnitude of LFEs of the 37102 family and display the results as follows. The magnitude range obtained by the EGF method is essentially the same as that of our method in the paper.

Using the EGF method to measure the magnitude of LFEs

We downloaded all waveform data from the Northern California Earthquake Data Center (NCEDC) using the Simple Waveform Client, corrected for instrument response. We performed our analysis using S waves identified on horizontal

components DP2 filtered in the 1–50 Hz frequency band. We used a multitaper method to deconvolve the EGF from the target event.

We first show the waveforms of LFEs and EGFs. The following figure shows the stacked LFE waveforms and 21401498 EGFs of the three stations FROB, CCRB and SMNB. Among them, the corresponding correlation coefficients between the stacked LFEs and EGFs are 0.1132, 0.2608 and 0.3987. The correlation coefficient between the unstacked LFEs and EGFs is 0.0692 at the FROB station, 0.2085 at the CCRB station, and 0.1976 at the SMNB station. The signal-to-noise ratios of the stacked LFEs is better, and the correlation with the EGF waveforms is higher.

Since our goal is to obtain the magnitude of a single unstacked low-frequency earthquake, we then estimate the moment magnitude of every single LFE in the 37102 family in our data using the EGF method. For all 83 LFEs in the 37102 family, the magnitude ranges from $9.4634 \times 10^9 N \cdot m$ to $8.5562 \times 10^{11} N \cdot m$. The average moment magnitude of LFEs in the 37102 family is $1.73 \times 10^{11} N \cdot m$. The magnitude range distribution of these LFEs is shown in the following figure. The magnitude of LFEs estimated through the EGF method is essentially consistent with the results obtained by our method.

In summary, both the method used in our article and the EGF method have their own advantages and disadvantages. However, we take into account the obvious shortcomings of the EGF method: the number of usable EGFs is too small, and the correlation coefficient is too low (far less than 0.9). We choose to use Chestler et al.²'s method, as presented in our paper. As Chestler et al.² wrote, "*While this calculation may seem crude, it is designed to obtain a robust estimate of moment by using signals that are coherent within each array even though the SNR is low.*", we consider this simple method a more suitable method for measuring the magnitude and stress drop of every single LFE. Additionally, we compare the measured magnitude of LFEs in the 37102 family with those obtained through the EGF method, and the results are similar, which further verifies that our method is reliable.

In addition, there are millions of LFEs in the Shelly (JGR, 2017) catalog. I do not think that the authors did millions of measurements, right? If so, what criteria did they use to select the LFEs in their analysis?

We thank the reviewer for this valuable comment. We actually did not measure all LFEs listed by Shelly (JGR, 2017)³. We selected the LFEs that had good correlation with the low-frequency seismic templates for measurement. Specifically, we selected LFEs with *meanc* values greater than or equal to 0.4 and *ccsum* values greater than or equal to 10 in Shelly's catalogue for source parameter estimation.

We added to I. 146-148: "*Using the LFE location data identified by Shelly, we select the LFEs with good correlation with the template ($meanc \geq 0.4$ and $ccsum \geq 10$) to estimate the source parameters.*" to clarify our selection criteria.

Finally, they used a relatively simple velocity model (Crust 1,0) and a quality factor of 700. There are more complicated 1D or 3D velocity model in this region, and the quality factor for seismic stations near faults might be much smaller. How would the choice of these parameters affect the results? This needs to be further explored/discussed in the paper.

We thank the reviewer for this suggestion. In our paper, we used Chestler et al.²'s method to estimate the moment magnitudes of LFEs, and this method is based on a 1D (two-layer) medium assumption. Because of the close horizontal distance between LFEs and stations, we suggest that the use of 3D models may not be necessary.

We agree that the quality factor is too large. Thus, we have changed the Q_s value from 700 to 200^{6,7}. We recalculated the moment magnitudes and stress drops of LFEs in the Parkfield area with a Q_s of 200 (Fig. 2). We chose this value based on the work from Abercrombie, R. E.⁶ and Hauksson et al.⁷. In Abercrombie, R. E.'s article, he concluded that "The San Andreas fault is confirmed to be a strongly attenuation zone with $Q_s \sim 80$ " and " $Q_{SW} \sim 200$ and $Q_{NE} \sim 100$ between ~ 200

and 5000 m depth for both P and S waves.”. Combined with the results from Hauksson et al.⁷ and considering that the depths of LFEs are deep (~20 km), we choose Q_s as 200, an estimate of the regional average attenuation factor.

We also added the discussion of Q_s value to l. 157-158: *“We have also studied the influence of different quality factors and filter frequency bands on the estimation results (Supplementary Fig. S6-8).”* and l. 467-476: *“The method of estimating moment magnitude and stress drop depends on the quality factor Q_s . Therefore, we also explore the difference between the results under different Q_s values (Supplementary Fig. S7 and S8). With the increase in the Q_s value, the smaller the average moment magnitude of LFEs obtained, the larger the average stress drop. However, the change is small, and the estimated 95% of the LFE source parameters are still within the range that we used to compare with the simulation results. Due to the close horizontal distance between the stations and LFEs, the change in the Q_s value has a limited effect on the estimated magnitude and stress drop. Thus, a value of 200 for Q_s is relatively reliable in estimating the range of moment magnitude and stress drop of LFEs.”*. The source parameter results using different Q_s values are shown in *Supplementary Figs. S7 and S8*.

- Equation 9 in page 27: might be better to change d^N_k and d^E_k to D^2_k and d^3_k since the two horizontal components are not aligned in the N-S and E-W direction.

We are very grateful to the reviewer for pointing out the inaccuracy of the formula. In addition, we have changed the expression of this formula.

We changed l. 449: *“ $M_0 \approx \frac{4\pi\beta_2^2 (\rho_1\beta_1 + \rho_2\beta_2) r}{2Rc} e^{\frac{rf\pi}{\beta Q_s}} \text{median}(d_k^N, d_k^E)$ ”* to

“ $M_0 \approx \frac{4\pi\beta_2^2 (\rho_1\beta_1 + \rho_2\beta_2) r}{2Rc} e^{\frac{rf\pi}{\beta Q_s}} \text{median}(d_k^2, d_k^3)$ ” and l. 436: *“(…) on components*

DP2 and DP3 filtered at 2-8 Hz.”.

In addition, the authors filtered the data only in the 2-8 Hz filter band. What is the justification for this range? Would the inclusion of high-frequency signal help in determining the corner frequency/moment?

We thank the reviewer for this comment. We did not estimate the source duration of LFEs in Parkfield. We estimated the seismic moment magnitude,

stress drop and average rupture radius of LFEs. In our paper, we directly use the source duration of LFEs estimated by Thomas et al.¹ (since we both use the same LFE data in the Parkfield area).

To address this question, we choose a filter frequency of 2-8 Hz, mainly referring to Chestler et al.². In their article (Methods section), they said “*We deconvolve our data to obtain displacement and filter it between 2 and 8 Hz.*”. Because our method estimates the magnitude by amplitude and the main frequency range of LFEs is mainly 1-10 Hz, we use this frequency band for estimation. We noticed that the limited frequency band actually has a significant impact on the LFE source *duration* measurement. It does not make much sense to use the filtered 2-8 Hz records to estimate the average duration of LFEs to obtain an average source duration of ~0.2 s since the corner frequency range of the frequency band is limited. However, for the estimation of moment magnitude, the frequency band of 2-8 Hz highlights the main frequency part of LFEs, which we think is reasonable.

We compared the effects of using the same LFE data to filter with frequency bands of 2-8 Hz and 2-50 Hz on the results of LFE source parameters. We show the results in *Supplementary Fig. S6*.

The moment magnitudes of LFEs estimated by filtering at 2-50 Hz are larger than those estimated by the 2-8 Hz data, but the stress drops are smaller. This is because when estimating the average rupture area, we simply give an average slip value. As the magnitude increases, while the slip value remains unchanged, the average rupture area obtained becomes larger. A larger rupture area results in a smaller estimated stress drop.

Using the data filtered by 2-8 Hz, the measured average M_0 of LFEs is 1.4110×10^{11} , and the average stress drop is $3.4327 \times 10^4 Pa$. Using the data filtered by 2-50 Hz, the measured average M_0 of LFEs is 2.2993×10^{11} , and the average stress drop is $2.4374 \times 10^4 Pa$. The two results are similar.

We added to l. 157-158: “*We have also studied the influence of different quality factors and filter frequency bands on the estimation results (Supplementary Fig. S6-8).*” and added *Supplementary Fig. S6*.

3. Pages 7 and 8, lines 133-134: please refer to the supp. Material here so that the readers know where to find additional information on how those measurements are made.

We are very thankful for this comment.

We added to l. 150: “*The specific estimation methods are provided in the Methods section.*”

Please specify how many measurements are made for the LFEs here. In addition, the authors only used the data from 2002 to 2010, while the catalog lasted much longer. Why not use the data after 2010?

We thank the reviewer for this comment. In our updated manuscript, we added the specific number of measurements we performed. We also downloaded LFE data from 2010 to 2016 (from NCEDC). Now, we have measured the source parameters of LFEs from 2002 to 2016, which is the time span in the LFE list given by Shelly (JGR, 2017)³. We updated *Fig. 2*.

We added to l. 148-150: *“We estimate the average moment magnitudes and stress drops of 6585 LFEs in every tremor family located in the Parkfield from 2002 to 2016 (Fig. 2).”*

4. Figure 1b shows three types of ruptures with different parameters. However, in the phase diagram in panel c, there are four types. Although it is not the focus of this study, I wonder if it would make sense to include the last type “Super-shear Rupture” in panel b as well (for completeness).

We are very grateful for this constructive suggestion. In addition, we have added super-shear rupture to *Fig. 1*.

We added the simulation parameters of this super-shear rupture to l. 102-103:

“In Fig. 1b, for the super-shear rupture, \hat{D}_c is 0.4, \hat{T}_e is 0.8; (...)”

5. Figure 2a: please add the fault map to show the San Andreas Fault as the background. In addition, it would be useful to add an inset to mark the study region in a larger map of CA or US (similar to the supp. Figure S5). Figure 2b and 2c: it is hard to see individual measurements. I wonder if it would make sense to add a separate panel to the right to show the histogram. There are some abnormally high or low values in both plots. What is the cause of these measurements?

We thank the reviewer for this suggestion. We added an inset according to the comment to *Fig. 2a*. In addition, we added a new panel to the right of *Fig. 2* to show the histograms of measurements.

There are some abnormally large and small values (moments and stress drops) in Fig. 2b, c. These estimated seismic moments and stress drops are abnormally large or small because the waveform amplitude of the LFEs corresponding to these data is too large or small. The reason for the abnormal amplitude of these data is not clear. We ruled out the possibility of instrument abnormalities or interference from simultaneous earthquakes, but no direct cause was found. We speculate that the large anomalies are due to the interference of some large low-frequency signals, while the small anomalies may be due to the existence of some small LFEs. Since we used the range where 90% of the source parameters of LFEs fall into to compare with the simulation results, these anomalous values did not affect the range of the source parameters.

6. Figure 3: are there any white stars near the right side of each panel? In figure

caption, change “pentagrams” to “stars”.

We are very grateful to the reviewers for pointing out the inappropriate expressions in our paper. Actually, there are only coloured stars near the right side of each panel. Because the edges of all stars are white and the stars are located very close to each other in the figure, it may lead the reader to think that there are some white stars near the right side. The stars of different colours are very close essentially because the source parameters of the SSARs under different rupture areas are actually very close.

We changed l. 165: “(...) *pentagrams*” to “(...) *stars*”

Additional minor comments:

Page 2, line 37-38: please try to cite a few references that have measured the stress-drop of LFEs (e.g., Thomas et al., GRL, 2016, which was cited later).

We thank the reviewer for this thorough reading and have added those references according to the reviewer’s suggestion.

Page 3, line 42-43: demonstrate their similarity?

We thank the reviewer for this comment and have changed the sentence to l. 43-44: “(...) *demonstrate that both of them are ruptures caused by slip on faults*”.

Page 3, line 55: change “Parkfield” to “the Parkfield-Cholame section of the San Andreas Fault (SAF), where tremor and LFEs have been recently observed (Nadeau and Guilhem, 2019; Shelly, 2017).”.

We are very thankful for this suggestion and have changed the phrase accordingly.

Page 3, line 57: add “Shelly (2010a, 2010b)” after the “characteristics of multiple LFEs”.

We thank the reviewer for this suggestion and have added the references.

Page 3, line 59-60: change the sentence to something like “connects with what controls ...” or “the underlying mechanism is the same for regular earthquakes”.

We thank the reviewer for this suggestion and have corrected the sentence.

Page 4, line 60: change ‘slow’ to slow (no need to quote it since it has been

mentioned before without quotation).

We thank the reviewer for this thorough reading and have corrected the error.

Page 4, line 76: perhaps start a new paragraph here.

We thank the reviewer for this suggestion and have started a new paragraph there.

Page 4, line 82: change “And the output” to “The output”. Can you specify what are the output (rupture process)?

We thank the reviewer for this suggestion and have specified the output of the rupture process. We changed |. 84-86: “And the output (...)” to *“The output is the rupture process, that is, the slip velocity and stress of the calculated area at each moment based on the given parameters.”*.

Page 4, line 86: change “100m, a spatial” to “100 m and a spatial”. Can you justify the choice of 100m radius?

We thank the reviewer for this suggestion and have corrected the error. A nucleation patch radius of 100 m was chosen because the average rupture radius of LFEs is ~100 m, and the nucleation patch radius used in the subsequent simulation of SSARs is also ~100 m. Therefore, we chose a nucleation patch radius of 100 m here for consistency with the subsequent simulations.

Page 6, lines 108-109: perhaps change the last sentence to something like this to make it clear: “Evidence of self-arresting events are observed for repeating earthquakes along the Parkfield section of the SAF14”.

We thank the reviewer for this constructive comment and have changed the sentence accordingly.

Page 7, Line 115-116: change “More extraordinarily” to “In particular” or something similar.

We thank the reviewer for this suggestion and have changed the phrase accordingly.

Page 7, line 122: change “Once” to “When” or “If”.

We thank the reviewer for this thorough reading and have changed “Once” to *“When (...)”*.

Page 7, line 127: change “shows that” to “confirms this inference” or “confirms this observation”.

We thank the reviewer for this suggestion and have changed the sentence accordingly.

Page 7, line 129: remove “generally” or change “the similarity” to “the general similarity”.

We thank the reviewer for this suggestion and have changed “generally” to “*the general similarity*”.

Page 8, lines 135-136: The last sentence “And we show...” can be removed by putting the reference to Fig. 2 to the previous sentence like “ from 2002 to 2010 (Fig. 2)”.

We thank the reviewer for this suggestion and have deleted the last sentence “And we show...”. In addition, we changed l. 148-150 as “*We estimate the average moment magnitudes and stress drops of 6585 LFEs in every tremor family located in the Parkfield from 2002 to 2016 (Fig. 2).*”.

Page 10, line 155: the sentence ends with “to account for” does not seem to be properly finished. Please correct.

We thank the reviewer for this thorough reading and have deleted “to account for”.

Page 10, lines 165-166: change “a long source duration earthquake” to “an earthquake with long-source duration”.

We thank the reviewer for this thorough reading and have changed the sentence accordingly.

Page 11, line 176: change “is required to accumulate to trigger” to “is needed to trigger”.

We thank the reviewer for this suggestion and have changed the sentence accordingly.

Page 11, line 185: it might useful to add the reference of Peng and Gomberg (NGEO, 2010), who noted similar scaling laws like in Ide et al. (Nature, 2007).

Peng, Z. and J. Gomberg (2010), An integrated perspective of the continuum between earthquakes and slow-slip phenomena, Nature Geosci., 3, 599–607, doi:10.1038/ngeo940.

We thank the reviewer for this suggestion and have added the references in our new version of the manuscript.

Page 11, line 188: change “slow self-arresting rupture” to “SSAR”.

We thank the reviewer for this suggestion and have changed the phrase accordingly.

Page 12, line 194: change “remain” to “keep”.

We thank the reviewer for this thorough reading and have changed the sentence accordingly.

Page 13, lines 199-200: change the sentence to “law, which is similar to regular earthquakes and is consistent with the newest studies¹⁹⁻²²”.

We thank the reviewer for this suggestion and have added the references.

Page 13, line 204: add some references for the moment magnitudes of VLFs.

We thank the reviewer for this suggestion and have added the references accordingly.

Page 13, lines 205-206: remove the words “The fact is that,”.

We thank the reviewer for this suggestion and have deleted the phrase.

Page 14, line 217: change “which are SSARs”, to “which are equivalent to the SSARs in our numerical simulations”.

We thank the reviewer for this constructive suggestion and have changed the sentence accordingly.

Page 14, line 218: change “stopping” to “rupture arrest”.

We thank the reviewer for this thorough reading and have changed the sentence accordingly.

Page 14, line 226: change “nearby” to “external”.

We thank the reviewer for this thorough reading and have changed the word accordingly.

Page 14, line 227: change “Love wave” to “body/surface waves from regional and

teleseismic earthquakes”.

We thank the reviewer for the comment and have changed the sentence accordingly.

Page 14, line 227: please add a few references on the tidal triggering and remotely triggered tremor at the Parkfield-Cholame section of the SAF.

Thomas, A.M., R. M. Nadeau, and R. Burgmann (2009) Tremor-tide correlations and near-lithostatic pore pressure on the deep San Andreas fault. *Nature*. doi:10.1038/nature08654

Peng, Z., J. E. Vidale, K. C. Creager, J. L. Rubinstein, J. Gomberg, and P. Bodin (2008), Strong tremor near Parkfield, CA excited by the 2002 Denali Fault earthquake, *Geophys. Res. Lett.*, 35, L23305, doi:10.1029/2008GL036080.

Peng, Z., J. E. Vidale, A. Wech, R. M. Nadeau and K. C. Creager (2009), Remote triggering of tremor along the San Andreas fault in central California, *J. Geophys. Res.*, 114, B00A06, doi:10.1029/2008JB006049.

Shelly, D. R., Z. Peng, D. P. Hill and C. Aiken (2011), Triggered creep as a possible mechanism for delayed dynamic triggering of tremor and earthquakes, *Nature Geosci.*, 4, 384–388, doi: 10.1038/ngeo1141.

Peng, Z., D. R. Shelly, and W. L. Ellsworth (2015), Delay dynamic triggering of deep tremor along the Parkfield-Cholame section of the San Andreas Fault following the 2014 M6.0 South Napa earthquake, *Geophys. Res. Lett.*, 42, 7916-7922, doi:10.1002/2015GL065277.

We thank the reviewer for this constructive suggestion and have added the above references.

Page 15, line 246: change “increasing the possibility of LFEs are SSARs in nature” to “furthering confirming our inference that LFEs are SSARs in nature”.

We thank the reviewer for this suggestion and have changed the sentence accordingly.

Page 15, line 251: change “And tremor” to “Hence, tremor”.

We thank the reviewer for this thorough reading and have changed the sentence accordingly.

Page 16, line 262: change “which characterized” to “which is characterized”.

We thank the reviewer for this suggestion and have changed the sentence accordingly.

Page 16, line 264: change “distance provide perfect explanation for the swarm-like LFEs” to “distance, provide a natural explanation for the swarm-like behaviour of LFEs”.

We are very thankful for the comment and have changed the sentence accordingly.

In Supplementary Material:

Page 22, line 381: please add some references after the first sentence (rupture problems).

We thank the reviewer for this thorough reading and have added the references.

Page 22, lines 381-382: change “And BIEM (Boundary Integral Equation Method)” to “Among them, the Boundary Integral Equation Method (BIEM)”.

We thank the reviewer for this suggestion and have changed the sentence accordingly.

Page 25, lines 449: change “characteristics. We” to “characteristics, we”.

We thank the reviewer for this suggestion and have changed the sentence accordingly.

Page 25, line 458: remove “since”.

We thank the reviewer for this thorough reading and have removed the word.

Page 27, line 488: add “of LFE source parameters” after “estimation”.

We thank the reviewer for this thorough reading and have added this phrase.

References

- 1 Thomas, A. M., Beroza, G. C. & Shelly, D. R. Constraints on the source parameters of low-frequency earthquakes on the San Andreas Fault. *Geophys Res Lett* **43**, 1464-1471 (2016).
- 2 Chestler, S. R. & Creager, K. C. Evidence for a scale-limited low-frequency earthquake source process. *J Geophys Res-Sol Ea* **122**, 3099-3114 (2017).
- 3 Shelly, D. R. A 15year catalog of more than 1 million low-frequency earthquakes: Tracking tremor and slip along the deep San Andreas Fault. *J Geophys Res-Sol Ea* **122**, 3739-3753 (2017).
- 4 Thurber, C., Roecker, S., Zhang, H., Baher, S. & Ellsworth, W. Fine-scale structure of the San Andreas fault zone and location of the SAFOD target earthquakes. *Geophys Res Lett* **31**, doi:Artn L12a0210.1029/2003gl019398 (2004).
- 5 Waldhauser, F. & Schaff, D. P. Large-scale relocation of two decades of Northern California seismicity using cross-correlation and double-difference methods. *J Geophys Res-Sol Ea* **113**, doi:Artn B0831110.1029/2007jb005479 (2008).
- 6 Abercrombie, R. E. Crustal attenuation and site effects at Parkfield, California. *J Geophys Res-Sol Ea* **105**, 6277-6286, doi:Doi 10.1029/1999jb900425 (2000).
- 7 Hauksson, E. & Shearer, P. M. Attenuation models (Q(P) and Q(S)) in three dimensions of the southern California crust: Inferred fluid saturation at seismogenic depths. *J Geophys Res-Sol Ea* **111**, doi:Artn B0530210.1029/2005jb003947 (2006).

Reviewer #2 (Remarks to the Author):

The paper presents a provocative interpretation of low frequency earthquakes based on the results of a numerical analysis of the rupture from which a phase diagram is proposed. In this diagram, the authors define a class of slow self-arresting ruptures and suggest that they correspond to SSEs in nature. This is an interesting study that potentially deserves publication, as it provides elements of interpretation in a discussion that is very active today. The authors limit themselves to a slip model with slip dependent friction. Of course, it is a simple physical model, and therefore it can be criticized in view of the complexity of nature, but it is the interest of this kind of approach to propose the simplest arguments and to show that they are compatible with the main characteristics of the observations.

Nevertheless, I have some questions and remarks that I think should be considered by the authors.

1. The question of the evolution towards instability of a surface with slip-dependent weakening has been treated in theoretical papers which describe the relations between initial perturbation, weakening rate and evolution time (for example Campillo, M. and I. Ionescu (1997), Initiation of antiplane shear instability under slip dependent friction, *Journal of Geophysical Research*, 102, 20363-20371.). Moreover, the evolution on a surface of finite size and the inhibition of the instability has been precisely studied (for example: Dascalu, C., I.R. Ionescu and M. Campillo (2000), Fault Finiteness and Initiation of Dynamic Shear Instability, *Earth and Planetary Science Letters*, 177, 163-1766., Uenishi and Rice (2003), Universal nucleation length for slip-weakening rupture instability under nonuniform fault loading *Journal of Geophysical Research* <https://doi.org/10.1029/2001JB001681>). I believe that these previous results are relevant for the understanding of the existence of the class of SSARs described in the paper.

We are very thankful to the reviewer for this valuable suggestion. Campillo and Ionescu^{1,2} investigated the initiation of dynamic antiplane slip instabilities of a slip-weakening fault in a homogeneous linear elastic medium that is prestressed uniformly up to the frictional threshold. In their analysis, an analytical expression for the slip was divided into two parts: solutions associated with positive eigenvalues (“dominant part”) and negative eigenvalues (“wave part”). It has been shown that the dominant part, characterized by exponential growth with time, controls the development of the instability, and the wave part becomes rapidly negligible when the instability develops. The effect of the slip-weakening rate on the duration of the nucleation phase and the critical fault length was evaluated.

We inferred that the nondimensional parameter $\beta = a \frac{(\mu_s - \mu_d)S}{GL_c}$ defined in their

article may represent the same meaning as the reciprocal of \hat{D}_c in our paper ($\beta = 1/\hat{D}_c$), where $2a$ is the fault length, μ_s and μ_d are respectively the static and dynamic friction coefficients, G is the shear rigidity of the fault and L_c is the critical slip. According to their analysis, the parameter β completely characterizes the stability. It quantitatively gives the limit between the stable ($\beta < \beta_0$, where β_0 is the smallest positive eigenvalue) and unstable ($\beta > \beta_0$) behaviours of the fault. They also gave the smallest eigenvalue as $\beta_0 = 1.15999388$. That is, $\frac{1}{\beta_0} = 0.8637265$, which corresponds to the position of the boundary between sub-Rayleigh and SSARs in our phase diagram.

Uenishi and Rice³ investigated the nucleation process of a slip-weakening fault under locally peaked “loading” stress. They proved that the nucleation length is independent of the shape of the loading stress distribution. The nucleation length depends only on the elastic modulus of the medium and the slip-weakening rate. The nucleation length form is the same as that of Campillo and Ionescu^{1,2}.

These works explain the existence of SSARs from the level of numerical analysis. Through spectral analysis, the boundary between the sub-Rayleigh rupture and SSARs is intrinsically determined by the first positive eigenvalue β_0 . We suggest that the phase diagram verifies this from the perspective of numerical simulation, and these results are consistent.

We thank the reviewer again for this suggestion and have added these works to I. 133-135: *“SSARs are similar to the rupture event investigated by previous works¹⁹⁻²¹, where the slip cannot reach D_c .”* and have added those mentioned papers to the references.

2. In this regime, (and in general in the nucleation zone) there is no propagative stress concentration and therefore the notion of rupture velocity is not defined.

This should be clarified because the distance-time passage used in the discussion on scaling seems to me very arbitrary.

We thank the reviewer for this constructive comment. We apologize for not explaining this point clearly in the manuscript. In our previous version of the manuscript, we used two nucleation methods. When studying the low-frequency earthquake (LFE) source parameters, we use a simple model that ruptures simultaneously in the nucleation patch. However, to explore the LFE scaling part, we adopted a nucleation patch with a given rupture velocity. This is because we consider that for an earthquake with a nucleation radius greater than 500 m, it is more severe for the points in the nucleation zone to rupture at the same time. Therefore, to be closer to the actual situation, we added a given rupture speed so that the rupture of the nucleation zone starts from the centre point and spreads to the entire nucleation zone at a constant rupture speed. For the simulations without rupture velocity, we give the whole nucleation zone a stress slightly greater than the shear strength (1.001 T_u) at 0 s so that the whole nucleation zone ruptures. For the simulations with rupture velocity, we give the stress, which is slightly larger than the shear strength at 0 s and propagates to the whole nucleation region at a given rupture velocity from the centre point of the nucleation patch. Therefore, the rupture starts at the centre point and extends outward to the whole nucleation patch at a given rupture velocity.

We agree that such settings were confusing and arbitrary in our previous manuscript. Therefore, to make the manuscript more consistent, we re-simulated the SSARs with different scales (the discussion on scaling part), all of which used the same nucleation method as before. That is, the rupture was triggered simultaneously in the nucleation zone without rupture velocity. We updated *Fig. 4* and changed the description accordingly.

3. The model is built with an initial forcing in a central zone and this point should be discussed precisely because it controls the duration, and potentially the final evolution (see point 1) for SSARs, that is in absence of unstable growth. I wonder what is the role of the initial conditions on the spectrum that is represented in Figure S4. Clarifying this issue seems very important in relation to the results showing a good quantitative correspondence between the model predictions and the observations. The authors should show that this success is not depending on an ad-hoc choice of the initial triggering.

We thank the reviewer for this constructive suggestion. We agree that the setting of the nucleation patch is important. We added the nucleation triggering method to l. 89-95: *“We assume that the initial shear stress inside the nucleation patch has reached the shear strength limit σ_u , while the initial stress outside the nucleation patch is σ_0 (since σ_r can be eliminated during the calculation, the*

initial stress in the nucleation patch is actually set to be T_u during simulations, while the initial stress outside the nucleation patch is T_e). At 0 s, we assign all points in the nucleation patch a stress slightly greater than the shear strength limit so that rupture occurs across the nucleation patch simultaneously.”

We also added numerical experiments to explore the connections between SSARs and the nucleation method. According to the previous studies on nucleation mentioned by the reviewer, we applied a stress triggering method similar to that of Uenishi and Rice³. We quantitatively studied the influence of the initial force on the SSAR results in three cases. The specific model settings are provided in a new part in *Supplementary Text*, and the simulation results are shown in *Supplementary Figs. S12-13* as well as *Supplementary Table S3*. The results show that under a different triggering method, the source characteristics of SSAR are still similar to those in the main text.

References

- 1 Campillo, M. & Ionescu, I. R. Initiation of antiplane shear instability under slip dependent friction. *J Geophys Res-Sol Ea* **102**, 20363-20371, doi:Doi 10.1029/97jb01508 (1997).
- 2 Dascalu, C., Ionescu, I. R. & Campillo, M. Fault finiteness and initiation of dynamic shear instability. *Earth Planet Sc Lett* **177**, 163-176, doi:Doi 10.1016/S0012-821x(00)00055-8 (2000).
- 3 Uenishi, K. & Rice, J. R. Universal nucleation length for slip-weakening rupture instability under nonuniform fault loading. *J Geophys Res-Sol Ea* **108**, doi:Artn 204210.1029/2001jb001681 (2003).

Reviewer #3 (Remarks to the Author):

General Comments

Low-frequency earthquakes (LFEs) are small events (magnitudes roughly between about 0.7 and 2.2) that occur during episodes of non-volcanic tremor. They are characterized by unusually long source durations, small slip rates, and small stress drops. Their source mechanics remain a mystery.

In this paper Wei et al. show that a standard 3D elastic fault model, with a slip-weakening friction law, can generate an LFE having these observed characteristics. Whether the model generates an LFE or a standard earthquake depends on the stress-state and friction parameters. The parameter space they explore is defined by two dimensionless parameters:

$$T_e = T_e / T_u \text{ and } D_c = D_c / (R_a T_u / \mu)$$

Where T_u is the breakdown stress, T_e is the dynamic stress drop, D_c is the critical slip distance in the slip-weakening friction law, and μ is the shear modulus of the medium.

In this (D_c, T_e) space LFE type ruptures occur at large values of D_c ($D_c > 0.85$) and for nearly the entire range of T_e , ($0 < T_e < 1$).

Beyond the general framework outlined above, I found this paper very difficult to follow. I do not question that they did the numerical simulations correctly, but the physical significance of the model parameters was not obvious to me. Granted, I am not an expert on seismic source modeling, but I am close enough to the field that if I don't understand, I suspect that the broader audience of Nature readers may have similar problems.

1. To be specific, the key element of the physics here is that rupture propagation in an LFE is arrested during the nucleation process. It seems important therefore that this nucleation process should be described in detail. For example, how do the authors nucleate a rupture in their model? Do they introduce a heterogeneity at the center of the nucleation patch?

We thank the reviewer for this valuable comment, and we added the nucleation method to l. 89-95: *"We assume that the initial shear stress inside the nucleation patch has reached the shear strength limit σ_u , while the initial stress outside the nucleation patch is σ_0 (since σ_r can be eliminated during the calculation, the initial stress in the nucleation patch is actually set to be T_u during simulations,*

while the initial stress outside the nucleation patch is T_e). At 0 s, we assign all points in the nucleation patch a stress slightly greater than the shear strength limit so that rupture occurs across the nucleation patch simultaneously.”.

We did not introduce heterogeneity at the centre of the nucleation patch. Before the rupture is triggered, the fault is still. To trigger the rupture, we gave all grid points in the nucleation patch a stress slightly greater than its shear strength, i.e., $1.001 T_u$, to start the rupture. Therefore, each point in the nucleation zone begins to rupture at 0 s. No additional stress is applied in the subsequent rupture process. The triggering of the rupture in our simulations is simple, without any special settings. Once the rupture occurs in the nucleation zone, whether it involves run-away earthquakes or SSARs completely depends on the frictional property of the fault.

As the reviewer suggested, nucleation is an essential process for SSARs. Thus, we also investigated the effect of another nucleation method on our results. And added this new part to *Supplementary Text*. The simulation results are shown in *Supplementary Figs. S12-13* as well as *Supplementary Table S3*, which shows that our result is not depending on the choice of the initial triggering.

Once nucleated, what physically determines whether the rupture will die before it reaches the critical radius R_c for run-away propagation thereby producing an LFE?

I do not find it surprising that LFE production occurs at the largest values of the scaled critical radius ($D_c > 0.85$). Generally speaking, rupture propagation is a competition between decreasing friction (slip-weakening in this case) and decreasing elastic driving stress that accompanies displacement on the fault. If the fault stiffness, which depends on the radius of the slipping patch, decreases faster than the resisting friction, then slip will be arrested. (Dieterich (1986) demonstrated that such considerations lead to a minimum critical radius R_c required for rupture propagation, which depends on D_c . For larger values of D_c , the friction resistance to slip decreases more slowly with displacement, and the critical radius for run-away propagation is larger. and for the same elastic parameters, it is more likely that the rupture will be arrested before it reaches R_c .

Dieterich, J. H. (1986), A model for the nucleation of earthquake slip. In: S. Das, J. Boatwright, and C.H. Scholz (Editors), *Earthquake Source Mechanics*. Am. Geophys. Union, M. Ewing Vol. 6, Geophys. Monogr., 37, 37-47.

We agree. For understanding SSARs, this explanation from the perspective of stress is completely correct, deep and insightful. As demonstrated by Andrews¹, energy is absorbed per unit area as the crack advances and the energy needed is proportional to \hat{D}_c . From the perspective of energy, our understanding is that the

energy accumulated in the nucleation zone is insufficient to maintain the continuous propagation of the rupture because a larger value of \hat{D}_c requires a larger rupture energy.

The rupture pattern of self-arrest represented by SSARs has been found by previous studies through numerical analysis. For details, please refer to the first question raised by reviewer #2. In our work, we connect SSARs with LFEs through comparing their source parameters. Our work suggests that LFEs and normal earthquakes can be explained by a unified framework through the phase diagram of rupture dynamics.

2. What is missing in this paper is some sense of whether the friction parameters that produce an LFE make physical sense. If the LFEs arrest at a radius near 200 meters as claimed here, what is the corresponding value of D_c ? How does this compare with values of D_c found for normal earthquakes? Why is D_c larger at the base of the seismogenic zone where LFEs are observed than at shallower depths where normal earthquakes occur?

This is a very good question involving the physical interpretation of our model. We deem this question very important and seek to illustrate this point. We draw the reviewer's attention to *Supplementary Table. S2*, in which we give the physical values of all parameters (including the value of D_c) used in numerical simulations.

According to the research of Waldhauser and Schaff², normal small earthquakes occur in the area near LFEs (distance < 3 km). According to our simulations, we set T_e to 1 MPa, V_p to 6 km/s, V_s to 3.464 km/s and density to 2670 kg/m³. To explain the physical meaning behind our simulation parameters, we added a physical model description of LFEs to l. 245-263: *"Here, we provide a physical model based on our simulation to illustrate how LFEs occur (Supplementary Fig. S10). We assume that both LFEs and normal earthquakes occur in the same area.*

We control the final rupture area to be 220 m in diameter, R_a of the SSARs is set to 110 m, and R_a of the normal small (run-away) earthquakes is set to 75 m.

According to the phase diagram and simulation results, the dimensionless \hat{T}_e range of the LFEs is 0.75~0.95, and that of the normal small earthquakes is 0.3 ~ 0.6. The dimensionless \hat{D}_c range of the LFEs is 0.85~0.99, and that of the normal small earthquakes is 0.3 ~ 0.75. We choose dimensionless (\hat{D}_c , \hat{T}_e) as (0.9, 0.85) for the SSARs and (0.5, 0.45) for the normal earthquakes. Therefore, the dimensions (D_c , T_u) of the SSARs are (3.6 mm, 1.17 MPa), and those of the

normal earthquake is (2.6 mm, 2.22 MPa). According to the simulation results, the SSARs are consistent with the source parameter range of the LFEs in Parkfield, while the normal earthquakes are not. The seismic moment magnitude of a normal earthquake is 2.6, which is consistent with the magnitude of normal small earthquakes recorded by the Northern California Earthquake Data Center (NCEDC). The T_u values of normal earthquakes and SSARs are very different. For

this normal earthquake with $T_u=2.22$ MPa, an additional triggering stress of 1.22

MPa is required. For the SSAR with $T_u=1.17$ MPa, an additional triggering stress of only 0.17 MPa is required.” and the model is shown in Supplementary Fig. S10.

The T_u values of normal earthquakes and SSARs are very different. For this normal earthquake with $T_u=2.22$ MPa, an additional stress of 1.22 MPa is required to trigger this normal earthquake. For an SSAR with $T_u=1.17$ MPa, an additional stress of only 0.17 MPa is required for it to occur. The difference in T_u corresponds to the difference between the media. According to the definition of

$T_u = \tau_u - \tau_f$, where τ_u is the upper yield point and τ_f is the sliding friction

level, T_u is proportional to the effective normal stress of the medium. We speculate that the T_u value of the area where SSARs/LFEs occur is smaller, which may be because the water in these areas causes the pore pressure to increase. Thus, the effective normal stress is small, as is T_u .

In terms of the dimensional value of D_c , D_c of the LFE/SSAR (3.6 mm) is not very different from that of a normal earthquake (2.6 mm). We would like to emphasize that $\widehat{D}_c = D_c / (R_a \frac{T_u}{\mu})$ is a composite dimensionless parameter, which

is related to the dimensional values of T_u , R_a , μ and D_c . If T_u of rocks with normal earthquakes is much larger, D_c of normal earthquakes may even be larger than that of LFEs/SSARs. As shown in the Supplementary Table S2, some ordinary earthquakes have a D_c value of 4 mm, while some SSARs have a D_c value of 2.58 mm.

According to the simulation results revealed in Fig. 2, SSARs with rupture diameters of 180 m and dimensional D_c of 2.4 mm and rupture diameters of 280 m and dimensional D_c of 4.3 mm are consistent with the LFE source parameters in Parkfield. Thus, we do not require a special (very large) D_c in our simulation of SSARs. Dimensional D_c does not have a clear physical explanation at present, and we admit that it is beyond our ability to explain its physical meaning and why D_c may be larger for SSARs. We believe that in the future, there will be a more experimentally based explanation of the physical meaning of D_c and that we will have a deeper understanding of the phase diagram and SSARs.

3. Finally, I am not convinced by the statement (line 186) “Accumulating new

evidences show that slow earthquakes follow a cubic moment-duration scaling law similar to regular earthquakes”. This is a prediction of their model which they would like to be true. However, as far as I can tell, the jury is still out on moment duration scaling as evidenced by the following (2020) JGR paper

Farge, Shapiro, and Frank, 2020. Moment-Duration Scaling of Low-Frequency Earthquakes in Guerrero, Mexico. JGR Solid Earth, 124 (8).

<https://doi.org/10.1029/2019JB019099>

“We find characteristic values of $M_0 \sim 3 \times 10^{12}$ N.m ($M_w \sim 2.3$) and $f_c \sim 3.0$ Hz with the corner frequency very weakly dependent on the seismic moment. This moment-duration scaling observed for Mexican LFEs is similar to one previously reported in Cascadia and is very different from the established one for regular earthquakes. This suggests that they could be generated by sources of nearly constant size with strongly varying intensities. LFEs do not exhibit the self-similarity characteristic of regular earthquakes, suggesting that the physical mechanisms at their origin could be intrinsically different.”

The Cascadia papers that find LFE duration is approximately independent of its magnitude are (Bostock et al., 2015; Thomas et al. 2016).

Bostock, M.G., Thomas, A. M., Savard G., Chuang L., & Rubin, A. M. (2015). Magnitudes and moment-duration scaling of low-frequency earthquakes beneath southern Vancouver Island, J. Geophys. Res. Solid Earth, 120, <https://doi.org/10.1002/2015JB012195>

Thomas, A. M., Beroza, G. C., and Shelly, D. R. (2016). Constraints on the source parameters of low-frequency earthquakes on the San Andreas Fault, Geophys. Res. Lett., 43, 1464– 1471, doi:10.1002/2015GL067173.

We apologize for the inappropriate description of the scaling of slow earthquakes and thank the reviewer for pointing it out. As noted, we agree that whether slow earthquakes follow cubic or proportional scaling is a controversial issue. We changed the words and sentences in this section to make them more objective. Evidence³⁻⁶ shows that slow earthquakes follow a scaling different from normal earthquakes. There is also evidence⁷⁻¹¹ that supports that they follow a cubic scaling similar to normal earthquakes.

For our manuscript, in the scaling part, we assume that slow earthquakes with different magnitudes are SSARs with different rupture sizes under the same frictional conditions. We would like to draw the reviewer’s attention to l. 205-208:

“We simulate SSARs with fixed $\hat{D}_c=0.9$ and $\hat{T}_e=0.5$, and the rupture patch radius varies from 200 m to 9 km. We maintain a dynamic stress drop T_e of 1

MPa and the same background velocity structure as in the previous simulations.”. We provided the possible scaling regulation of LFEs based on the above assumptions. We changed the description of our measured scaling characteristics of SSARs, l. 216-217: *“Our results suggest that the SSARs follow a cubic scaling law that is similar to regular earthquakes and is consistent with some of the studies²⁶⁻²⁹.”*. However, the situation is more complicated for LFEs in the real world. They may be SSARs with different rupture sizes and different types (different D_c and T_u values).

We added the illustration of this point to l. 225-230: *“Note also that the similar slope result is based on the assumption that slow earthquakes of different sizes occur under the same frictional conditions with the same \hat{T}_e and \hat{D}_c . Therefore, with more observations and experiments uncovering the stress and slip-weakening conditions of LFEs in nature, we may find a more accurate and universal scaling law of LFEs.”*.

We also performed more numerical experiments to explain the phenomenon that the source duration of LFEs is independent of the moment magnitude. We added to l. 285-297: *“Another unusual characteristic of LFEs reported is that there are some LFEs whose source durations are approximately independent of their seismic moment magnitudes^{8,49-51}. This feature may also be explained by SSARs.*

We simulate SSARs with the same \hat{D}_c value of 0.88, the same rupture patch diameter of 240 m and \hat{T}_e ranging from 0.1 to 0.9. The results show that simulated SSARs have the same source duration but different seismic moments (Supplementary Fig. S11). Their moment magnitudes range from 1.47 to 2.11, and the source durations are both 0.22 s. This is because under the same rupture patch size (nucleation patch size), \hat{D}_c mainly controls the slip rate and propagation method of SSAR, while \hat{T}_e determines the magnitude of absolute shear strength (T_u). Under the same \hat{D}_c and rupture patch size, the durations of SSARs are the same, but the moment magnitudes are different due to the difference in \hat{T}_e . Therefore, the SSAR is a possible physical model of LFE and explains most of the source features of LFEs.”.

Specific line-by-line comments

15. and 33. Is it established that volcanic tremor consists entirely of LFEs?

We thank the reviewer for this question and have changed the term on l. 16 and

33: “(...) tectonic tremor”.

42. Waveform correlations between LFEs and regular earthquakes suggest that they share a similar source process. Is this true?

We thank the reviewer for pointing out this inaccurate expression and have changed the sentence in l. 41-44: *“However, despite the differences between these two catalogues of earthquakes, waveform correlations between LFEs and regular earthquakes demonstrate that both are ruptures caused by slip on faults².”*.

52. how is the nucleation zone defined?

We thank the reviewer for pointing out this point and have added explanations to

l. 89-93: *“We assume that the initial shear stress inside the nucleation patch has reached the shear strength limit σ_u , while the initial stress outside the nucleation patch is σ_0 (since σ_r can be eliminated during the calculation, the initial stress in the nucleation patch is actually set to be T_u during simulations, while the initial stress outside the nucleation patch is T_e).”* and |. 81-82: *“(…) R_a is the effective radius of the nucleation asperity, which is the radius of the assumed area for the elevated stress to trigger rupture”*.

116. How do you set the patch in the simulation to initiate the earthquake.

We thank the reviewer for this question and have added the following to l. 93-95: *“At 0 s, we assign all points in the nucleation patch a stress slightly greater than the shear strength limit so that rupture occurs across the nucleation patch simultaneously.”*.

137. Are these estimates from your model or the data?

We thank the reviewer for this question; these estimations are from data. We measured the LFE average moment magnitude and stress drop in the Parkfield area, as shown in *Fig. 2*.

139. where does the number 240 m for the rupture patch diameters come from?

We thank the reviewer for this question. The 240 m diameter for the rupture patch is based on our estimation of the average rupture patch diameter of LFEs

in Parkfield from real data. We would like to draw the reviewer's attention to I. 151-154: "Our results show that the LFEs moment magnitudes are between 0.7 and 2.2, the stress drops range from 8.1×10^3 Pa to 1.0×10^5 Pa and the rupture patch diameters are ~ 240 m (Fig. 2), which are close to values estimated in previous works (Supplementary Table. S1)." and J. 168-173: "With the average rupture area diameter for LFEs being around 200m, we simulate all three types of earthquakes with rupture patch diameters set at 180m, 220m, 260m and 300m (super-shear ruptures are not relevant). We use four rupture patch diameters during simulations to minimize the impact of the inaccuracy of estimated LFEs average diameter from seismic data on the comparison results."

Summary

In summary the major claim in this paper is that it is possible to produce LFE events using a standard earthquake model under special conditions of stress and friction. I am not convinced that this demonstration is of sufficient general interest to warrant publication in Nature, especially in view of their limited physical interpretation of the parameters in their model and the controversial nature of many of the observations offered in support. More suitable venues might be the Journal of the Seismological Society of America or the Journal of Geophysical Research which are aimed at a more specialized audience that could better assess the importance of this paper in the context of the extensive research on Low Frequency Earthquakes.

We are very grateful for the valuable comments and suggestions raised by Reviewer #3. We kindly disagree that there exists controversial nature of the observations and our simulations. As illustrated in the response, the independence of duration-moment (non-self-similar) characteristic of LFEs may also be explained by our model. After all, the goal of our model is to provide a possible framework about why LFEs are slow and inefficient in radiate seismic waves compared with normal earthquakes, which is basically the most important feature of LFEs. We believe the model could explain more characteristics observed from real data. And we hope the revision and response have addressed your concerns properly.

References

- 1 Andrews, D. J. Rupture Velocity of Plane Strain Shear Cracks. *J Geophys Res* **81**, 5679-5687, doi:DOI 10.1029/JB081i032p05679 (1976).
- 2 Waldhauser, F. & Schaff, D. P. Large-scale relocation of two decades of Northern California seismicity using cross-correlation and double-difference methods. *J Geophys Res-Sol Ea* **113**, doi:Artn B0831110.1029/2007jb005479 (2008).
- 3 Ide, S., Beroza, G. C., Shelly, D. R. & Uchide, T. A scaling law for slow earthquakes. *Nature* **447**, 76-79 (2007).
- 4 Bostock, M. G., Thomas, A. M., Savard, G., Chuang, L. & Rubin, A. M. Magnitudes and moment-duration scaling of low-frequency earthquakes beneath southern Vancouver Island. *J Geophys Res-Sol Ea* **120**, 6329-6350 (2015).
- 5 Peng, Z. G. & Gomberg, J. An integrated perspective of the continuum between earthquakes and slow-slip phenomena. *Nat Geosci* **3**, 599-607, doi:10.1038/Ngeo940 (2010).
- 6 Farge, G., Shapiro, N. M. & Frank, W. B. Moment-Duration Scaling of Low-Frequency Earthquakes in Guerrero, Mexico. *J Geophys Res-Sol Ea* **125**, doi:ARTN e2019JB01909910.1029/2019JB019099 (2020).
- 7 Dal Zilio, L., Lapusta, N. & Avouac, J. P. Unraveling Scaling Properties of Slow-Slip Events. *Geophys Res Lett* **47**, doi:ARTN e2020GL08747710.1029/2020GL087477 (2020).
- 8 Frank, W. B. & Brodsky, E. E. Daily measurement of slow slip from low-frequency earthquakes is consistent with ordinary earthquake scaling. *Sci Adv* **5**, doi:ARTN eaaw938610.1126/sciadv.aaw9386 (2019).
- 9 Michel, S., Gualandi, A. & Avouac, J. P. Similar scaling laws for earthquakes and Cascadia slow-slip events. *Nature* **574**, 522-+, doi:10.1038/s41586-019-1673-6 (2019).
- 10 Takagi, R., Uchida, N. & Obara, K. Along-Strike Variation and Migration of Long-Term Slow Slip Events in the Western Nankai Subduction Zone, Japan. *J Geophys Res-Sol Ea* **124**, 3853-3880, doi:10.1029/2018jb016738 (2019).
- 11 Supino, M. *et al.* Self-similarity of low-frequency earthquakes. *Sci Rep-Uk* **10**, doi:ARTN 652310.1038/s41598-020-63584-6 (2020).

We thank the referee for the comments, which have helped greatly improve our manuscript.

We hope that our response has suitably addressed the points raised in his/her review and that he/she is now convinced of the novelty and significance of our contribution.

REVIEWERS' COMMENTS

Reviewer #1 (Remarks to the Author):

The revised version has largely addressed most of my (as well as other reviewers' comments). The manuscript is almost ready to be accepted. Below are a few minor comments/suggestions that the authors may consider to improve it further:

1. My understanding is that for Nature Comm. one can include more than 4 figures. If so, perhaps consider moving a few figures back from the supp. material to the main figure (like Figure S10).

2. One advantage of the numerical modeling is that it can provide guidance for future field observation. I wonder if the authors can add a few sentences near the end of the discussion to describe what are possible new predictions from their numerical simulations that have and have not been observed in the field. This could potentially help to guide future seismological studies. Please ignore if this does not apply.

3. Lines 231-232: There is a main title "Discussion" and a subtitle "discussion and implication for slow earthquake". However, since there is only one subtitle, there is no need to have both. Perhaps replace "Discussion" with ""Discussion and implication for slow earthquake"

4. Reference-line 506: change "15yr" to "15 yr". In addition, please make sure that all the reference formats (lower/upper cases) are consistent.

5. The authors have improved Figure 2a, but it still does not look very informative when comparing with their Figure S5. I would suggest that the authors replace Figure 2a with Figure S5 (including the cross-section plot), and reorganize the rest of Figure 2.

6. The Data availability statement is too short and does not contain enough useful information for other readers to reproduce their work. I would suggest something like this: "The HRSN borehole seismic data (network code BP) can be downloaded from the NCEDC (<http://ncedc.org/>). The tremor/LFE catalog is obtained from Data S1 of Shelly et al. (2017).

7. The Code availability statement does not contain useful information for others to reproduce the work. The words "reasonable request" give authors an opportunity to reject readers' request. From what I can tell, there are at least two types of codes that are involved in this study. The first is the numerical modeling code, and the second is the code for measuring the source properties of LFEs. It would be great if the authors can put relevant to github to other places that readers can access freely. If not possible, I would suggest that the authors stated clearly what are the names of the several codes used in this study, and change "on reasonable request" to "upon request" or something similar.

Reviewer #2 (Remarks to the Author):

The authors have provided the necessary clarifications for a good understanding of the model they propose. I consider that the paper can be published.

Reviewer #3 (Remarks to the Author):

The authors have responded in detail to my comments. I am satisfied with their changes and recommend publication of the revised manuscript.

Reviewers' comments

Reviewer #1 (Remarks to the Author):

The revised version has largely addressed most of my (as well as other reviewers' comments). The manuscript is almost ready to be accepted. Below are a few minor comments/suggestions that the authors may consider to improve it further:

1. My understanding is that for Nature Comm. one can include more than 4 figures. If so, perhaps consider moving a few figures back from the supp. material to the main figure (like Figure S10).

2. One advantage of the numerical modeling is that it can provide guidance for future field observation. I wonder if the authors can add a few sentences near the end of the discussion to describe what are possible new predictions from their numerical simulations that have and have not been observed in the field. This could potentially help to guide future seismological studies. Please ignore if this does not apply.

3. Lines 231-232: There is a main title "Discussion" and a subtitle "discussion and implication for slow earthquake". However, since there is only one subtitle, there is no need to have both. Perhaps replace "Discussion" with ""Discussion and implication for slow earthquake"

4. Reference-line 506: change "15yr" to "15 yr". In addition, please make sure that all the reference formats (lower/upper cases) are consistent.

5. The authors have improved Figure 2a, but it still does not look very informative when comparing with their Figure S5. I would suggest that the authors replace Figure 2a with Figure S5 (including the cross-section plot), and reorganize the rest of Figure 2.

6. The Data availability statement is too short and does not contain enough useful information for other readers to reproduce their work. I would suggest something like this: "The HRSN borehole seismic data (network code BP) can be downloaded from the NCEDC (<http://ncedc.org/>). The tremor/LFE catalog is obtained from Data S1 of Shelly et al. (2017).

7. The Code availability statement does not contain useful information for others to reproduce the work. The words "reasonable request" give authors an opportunity to reject readers' request. From what I can tell, there are at least two types of codes that are involved in this study. The first is the numerical modeling code, and the second is the code for measuring the source properties of LFEs. It would be great if the authors can put relevant to github to other places that readers can access freely. If not possible, I would suggest that the authors stated clearly what are the names of the several codes used in this study, and change "on reasonable request" to "upon request" or something similar.

Reviewer #2 (Remarks to the Author):

The authors have provided the necessary clarifications for a good understanding of the model they propose.

I consider that the paper can be published.

Reviewer #3 (Remarks to the Author):

The authors have responded in detail to my comments. I am satisfied with their changes and recommend publication of the revised manuscript.

We appreciate the positive comments, acceptance, and agreement with the revised manuscript. We have updated our paper according to the suggestions of reviewer #1.

The answers to the reviewers are shown in blue.

Modifications of the manuscript in orange

Reviewer #1 (Remarks to the Author):

The revised version has largely addressed most of my (as well as other reviewers' comments). The manuscript is almost ready to be accepted. Below are a few minor comments/suggestions that the authors may consider to improve it further:

1. My understanding is that for Nature Comm. one can include more than 4 figures. If so, perhaps consider moving a few figures back from the supp. material to the main figure (like Figure S10).

We thank the reviewer for this suggestion. We consider that what Figure S10 shows is a relatively simple model, so we still choose to put it in the supplementary materials.

2. One advantage of the numerical modeling is that it can provide guidance for future field observation. I wonder if the authors can add a few sentences near the end of the discussion to describe what are possible new predictions from their numerical simulations that have and have not been observed in the field. This could potentially help to guide future seismological studies. Please ignore if this does not apply.

We thank the reviewer for this suggestion. Based on our model, we explained the main source characteristics of LFEs. Our model provides predictions in the scaling property of slow earthquakes, that is, our model may also explain the main characteristics of very low-frequency earthquakes (VLFs) source parameters, and predicts that the scaling property of slow earthquakes may be similar to ordinary earthquakes. As for the prediction of LFEs, we previously found that our model will produce LFEs with the same duration but very different moment magnitudes. However, according to the comments of reviewer #3, and we also searched more relevant works and found that this phenomenon has been observed. At present we have no more findings and predictions, we will continue to work in this direction. Thank you very much for your suggestion.

3. Lines 231-232: There is a main title "Discussion" and a subtitle "discussion and implication for slow earthquake". However, since there is only one subtitle, there is no need to have both. Perhaps replace "Discussion" with ""Discussion and implication for slow earthquake"

We thank the reviewer for this suggestion. According to the requirement of Nature

Communications, the title of discussion can only be "Discussion". Thus, in order to avoid repetition, we deleted the subtitle "*Discuss and implication for slow earthquakes*".

4. Reference-line 506: change "15yr" to "15 yr". In addition, please make sure that all the reference formats (lower/upper cases) are consistent.

We thank the reviewer for this comment and changed "15yr" to "15 yr".

5. The authors have improved Figure 2a, but it still does not look very informative when comparing with their Figure S5. I would suggest that the authors replace Figure 2a with Figure S5 (including the cross-section plot), and reorganize the rest of Figure 2.

We thank the reviewer for this suggestion. In order to enrich Figure 2a, we have added the magnitude of the LFEs we measured with colorful dots.

6. The Data availability statement is too short and does not contain enough useful information for other readers to reproduce their work. I would suggest something like this: "The HRSN borehole seismic data (network code BP) can be downloaded from the NCEDC (<http://ncedc.org/>). The tremor/LFE catalog is obtained from Data S1 of Shelly et al. (2017).

We appreciate this comment. We have modified the Data availability statement according to this suggestion.

7. The Code availability statement does not contain useful information for others to reproduce the work. The words "reasonable request" give authors an opportunity to reject readers' request. From what I can tell, there are at least two types of codes that are involved in this study. The first is the numerical modeling code, and the second is the code for measuring the source properties of LFEs. It would be great if the authors can put relevant to github to other places that readers can access freely. If not possible, I would suggest that the authors stated clearly what are the names of the several codes used in this study, and change "on reasonable request" to "upon request" or something similar.

We thank the reviewer for this comment. We have uploaded the codes we used to git-hub. And we have updated the Code availability statement according to the suggestion.

We appreciate the valuable comments and suggestions from Prof Peng. Thanks again for the considerable effort he put into reviewing our manuscript.

Reviewer #2 (Remarks to the Author):

The authors have provided the necessary clarifications for a good understanding of the model they propose.

I consider that the paper can be published.

We appreciate the reviewer for the affirmation of our work. We are very pleased to hear that our revised paper addresses all raised concerns and would like to thank again the reviewer for the constructive feedback.

Reviewer #3 (Remarks to the Author):

The authors have responded in detail to my comments. I am satisfied with their changes and recommend publication of the revised manuscript.

We thank the reviewer for accepting and agreeing to our changes and extensions of the manuscript. We are pleased to hear that our revised version addresses all raised concerns and would like to thank again the reviewer for the constructive feedback.